# GROKKED MODELS ARE BETTER UNLEARNERS

## ABSTRACT

*Grokking*—delayed generalization that emerges well after a model has fit the training data—has been linked to robustness and representation quality. We ask whether this training regime also helps with *machine unlearning*, i.e., removing the influence of specified data without full retraining. We compare applying standard unlearning methods *before* versus *after* the grokking transition across vision (CNNs/ResNets on CIFAR, SVHN and ImageNet) and language (a transformer on a TOFU-style setup). Starting from grokked checkpoints consistently yields (i) more **efficient forgetting** (fewer updates to reach a target forget level), (ii) **less collateral damage** (smaller drops on retained and test performance), and (iii) **more stable updates** across seeds, relative to early-stopped counterparts under identical unlearning algorithms. Analyses of features and curvature further suggest that post-grokking models learn *more modular representations* with reduced gradient alignment between forget and retain subsets, which facilitates selective forgetting. Our results highlight **when** a model is trained (pre- vs. post-grokking) as an orthogonal lever to **how** unlearning is performed, providing a practical recipe to improve existing unlearning methods without altering their algorithms.

## 1 INTRODUCTION

The rise of machine learning has brought transformative advancements across domains, yet this progress comes with growing concerns about data privacy, regulatory compliance (e.g., GDPR, CCPA), and the "right to be forgotten." Traditional machine learning models stubbornly retain information from their training data, making selective data removal challenging without costly retraining. This has made machine unlearning—the process of removing specific data influences from trained models—a critical research area with significant computational and performance challenges.

A key challenge in machine unlearning is that existing methods often degrades model performance on retained data or requires extensive computational resources. The effectiveness of unlearning depends heavily on the internal structure and representational quality of the trained model. Models with better-organized, more disentangled representations should theoretically enable more selective and stable forgetting. This raises a fundamental question: **what training dynamics produce models that are inherently better suited for unlearning?**

Recent discoveries in deep learning provide a surprising answer. The phenomenon of **grokking** (Power et al., 2022)—where models achieve delayed but strong generalization long after overfitting—challenges traditional training paradigms. Grokked models demonstrate superior robustness (Humayun et al., 2024) and generalization (Liu et al., 2022) compared to early-stopped counterparts, suggesting they develop fundamentally different internal representations.

This connection between representation quality and unlearning effectiveness leads to an intriguing paradox. On one hand, grokked models develop better generalization and more robust representations, which should theoretically facilitate selective forgetting by creating more disentangled knowledge structures. On the other hand, grokking requires extensive training on the data, potentially causing models to "remember" information more deeply, making unlearning more difficult. This raises a critical question: which effect dominates in practice?

We resolve this paradox by demonstrating that **the representational benefits of grokking outweigh the memorization concerns**. As illustrated in Figure 1, while extended training before grokking indeed makes unlearning progressively more difficult—with unlearning accuracy increasing and tracking closely to retain accuracy due to entangled representations—the grokking transition funda-

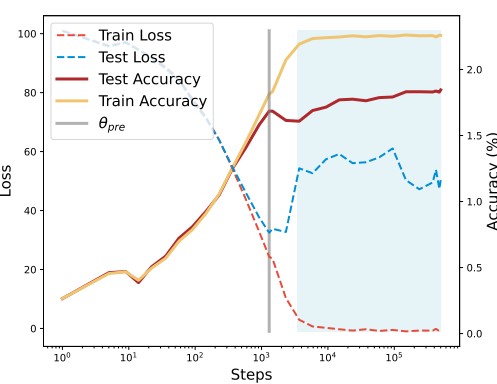 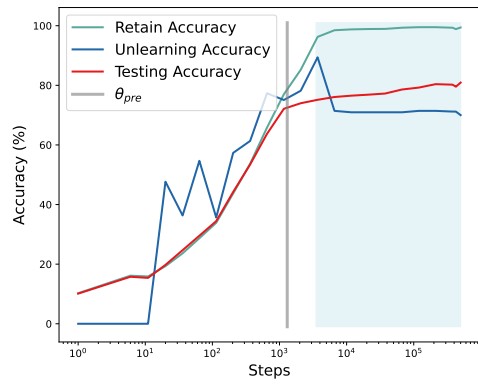

(a) Training dynamics showing grokking phenomenon

(b) Unlearning across multiple training checkpoints

Figure 1: **Grokking Enables Superior Machine Unlearning. (a) Training Dynamics:** ResNet training on CIFAR-10 showing grokking transition from conventional early stopping at $\theta_{\text{pre}}$ (pink region) through overfitting to delayed generalization at $\theta_{\text{grok}}$ (blue region). **(b) Unlearning Performance:** Gradient ascent unlearning effectiveness across training checkpoints. Higher UA indicates worse unlearning (model remembers what it should forget). Pre-grokking shows concerning upward UA trend with high volatility and poor selectivity (UA ≈ RA), indicating entangled representations where unlearning algorithms cannot distinguish forget from retain data. Post-grokking shows dramatic improvement: UA drops significantly below RA and stabilizes, demonstrating selective forgetting capability. This UA-RA separation reveals that grokking reorganizes representations into more modular, disentangled structures enabling precise unlearning operations.

mentally changes this dynamic. After grokking, models exhibit dramatically improved unlearning selectivity: unlearning accuracy drops significantly below retain accuracy and stabilizes, enabling precise data removal while preserving useful knowledge.

Through comprehensive experiments across vision and language domains, we show that grokked models consistently exhibit superior unlearning capabilities. When subjected to state-of-the-art unlearning algorithms—gradient ascent, SCRUB, Fisher forgetting, and fine-tuning—grokked models achieve more efficient data removal while better preserving performance on remaining data and maintaining enhanced robustness. Our findings are striking: grokked models achieve 6-8% better unlearning effectiveness while maintaining 10-20% higher performance on retained data compared to non-grokked counterparts, making privacy-preserving machine learning more practical.

Our analysis reveals that grokking fundamentally restructures internal representations in ways that facilitate selective forgetting with minimal collateral damage. This suggests that the training dynamics leading to grokking can be strategically leveraged to develop more practical privacy-preserving machine learning systems. While Zhao et al. (2024) identify strong entanglement between forget and retain sets as a key difficulty in unlearning, our work complements this by showing that grokking naturally reduces such entanglement by reorganizing representations and flattening the loss landscape. This, in turn, allows existing unlearning algorithms to perform more effectively without modification.

Our contributions are as follows:

1. We establish the first systematic connection between grokking and machine unlearning, resolving the apparent paradox between extensive training and effective forgetting.

2. We provide comprehensive empirical evidence across vision (CNNs/ResNets on CIFAR) and language (transformers on TOFU) domains, demonstrating that grokked models exhibit superior unlearning capabilities across diverse algorithms (gradient ascent, SCRUB, Fisher forgetting, fine-tuning).

3. We reveal the mechanistic basis for grokking's unlearning advantages through gradient correlation and local complexity analyses, showing that grokked models develop more orthogonal optimization pathways and simpler representational structures that facilitate selective forgetting.

4. We demonstrate that grokked models provide a practical training paradigm for privacy-preserving applications, achieving more efficient data removal while maintaining enhanced robustness and performance retention without requiring new unlearning algorithms.

## 2 MOTIVATION AND RESEARCH SCOPE

This work establishes a fundamental understanding of how model representations affect unlearning effectiveness, using grokking as an experimental lens rather than a practical recommendation. We investigate which representational properties enable effective unlearning and why they matter.

**Conceptual Contribution vs. Practical Recommendation** The grokking literature (Power et al., 2022; Liu et al., 2022) studies delayed generalization without advocating for thousand-epoch training in practice. Similarly, we use pre-grokking versus post-grokking comparisons to isolate specific representational properties—modularity, gradient orthogonality, and loss landscape flatness—that facilitate selective forgetting. While grokking incurs substantial training overhead (5-6× longer training), our goal is to identify the underlying mechanisms that enable superior unlearning, not to prescribe extended training as a practical solution. Our experiments with Sharpness-Aware Minimization (SAM) (Foret et al., 2020) demonstrate that some benefits can be achieved without full grokking, suggesting more efficient paths to these advantageous properties.

**Research in Context** Our approach follows an established pattern in deep learning research: identifying beneficial properties through idealized conditions, then developing practical approximations. Research on flat minima (Hochreiter & Schmidhuber, 1997) led to optimizers like SAM (Foret et al., 2020), while studies of representation disentanglement (Bengio et al., 2013) informed methods for learning more structured features (Chen et al., 2018). Similarly, our identification of representational characteristics that enable effective unlearning lays groundwork for future research on inducing these properties efficiently without the computational burden of extended training.

**Key Questions** This paper addresses three questions: (1) Do grokked models exhibit superior unlearning capabilities? (2) What specific representational properties explain these differences? (3) Can these beneficial properties be partially induced through more efficient methods? By answering these questions, we advance the theoretical understanding of machine unlearning while providing insights for developing more practical approaches in the future.

## 3 BACKGROUND AND RELATED WORKS

### 3.1 GROKKING: DELAYED GENERALIZATION IN DEEP LEARNING

**Discovery and properties.** Grokking refers to a training regime where models first overfit, then after prolonged stagnation, undergo sharp transitions to strong generalization (Power et al., 2022). Originally observed in modular arithmetic with Transformers, grokking has since been documented across diverse tasks—group theory (Chughtai et al., 2023), image classification (Liu et al., 2022)—and architectures, suggesting a fundamental training dynamic robust to optimization choices (Gromov, 2023).

**Theoretical interpretations.** Multiple theories explain grokking through implicit bias and phase transitions. Lyu et al. (2023) formalize a transition from "lazy" (kernel-like) to "rich" feature-learning regimes, while Zhu et al. (2024) identify data-dependent thresholds for reliable grokking. These accounts suggest discontinuous shifts in representation space, with gradient descent eventually preferring simpler, generalizable solutions over complex memorizing ones (Davies et al., 2023).

**Mechanistic insights.** Interpretability studies reveal network reorganization at grokking transitions. Nanda et al. (2023) show Transformers transition from distributed co-adaptation to modular subcircuits implementing algorithmic solutions. This distributed-to-modular shift involves competition between dense memorizing and sparse generalizing circuits (Merrill et al., 2023; Varma et al., 2023), with landscape changes toward flatter minima (Notsawo Jr et al., 2023). Crucially, post-grokking models exhibit more structured, modular representations (Humayun et al., 2024; Furuta et al., 2024)—precisely the type of organization we hypothesize enables effective selective forgetting.

### 3.2 MACHINE UNLEARNING: SELECTIVE DATA REMOVAL

Machine unlearning aims to remove the influence of a designated subset $\mathcal{D}_{\text{forget}} \subset \mathcal{D}$ from a model's parameters, producing behavior indistinguishable from training on $\mathcal{D}_{\text{retain}} = \mathcal{D} \setminus \mathcal{D}_{\text{forget}}$. Applications range from class unlearning (removing entire categories) to sample unlearning (specific identities or documents) (Choi & Na, 2023; Poppi et al., 2024).

**Exact vs. approximate unlearning.** Exact unlearning via retraining provides strongest guarantees but is computationally prohibitive. Approximate methods seek functional equivalence to retraining while avoiding full computational cost (Bourtoule et al., 2021; Izzo et al., 2021).

### 3.2.1 APPROXIMATE UNLEARNING METHODS

**Gradient-based methods** apply gradient ascent on $\mathcal{D}_{\text{forget}}$: $w \leftarrow w + \eta \nabla_w \mathcal{L}_{\text{forget}}(w)$, but often harm $\mathcal{D}_{\text{retain}}$ performance. Enhanced variants like $\nabla \tau$ (Trippa et al., 2024) interleave ascent on forget data with descent on retain data.

**Influence-based methods** estimate parameter shifts from data removal: $\Delta w \approx -\frac{1}{n} H^{-1} \nabla_w \ell(z; w)$, where $H$ is the training loss Hessian (Koh & Liang, 2017; Izzo et al., 2021). Practical implementations use structured approximations due to computational constraints.

**Fisher forgetting** injects curvature-guided noise aligned to Fisher information on $\mathcal{D}_{\text{forget}}$, randomizing sensitive parameters while preserving others (Golatkar et al., 2020).

**Distillation-based methods** train students to match teachers on $\mathcal{D}_{\text{retain}}$ while diverging on $\mathcal{D}_{\text{forget}}$. SCRUB uses negative-KL divergence (Kurmanji et al., 2023), while Bad Teacher employs dual teachers for controlled knowledge transfer (Chundawat et al., 2023).

**LLM approaches** typically use constrained fine-tuning with KL anchoring. Methods include Negative Preference Optimization (NPO) (Zhang et al., 2024) and Representation Misdirection (RMU) (Li et al., 2024). Evaluation benchmarks like TOFU (Maini et al., 2024a) reveal that current methods fail to match retraining baselines, highlighting the need for improved approaches.

### 3.3 EVALUATING MACHINE UNLEARNING

Evaluating unlearning requires assessing forgetting effectiveness, retention of useful performance, privacy verification, and efficiency.

**Core metrics.** Standard measures include Unlearning Accuracy (UA) on $\mathcal{D}_{\text{forget}}$ (lower indicates better forgetting), Retain Accuracy (RA) on $\mathcal{D}_{\text{retain}}$ (higher indicates better preservation), and Test Accuracy (TA) on held-out data. Relative metrics like Retain Retention (RR) compare against retrained baselines.

**Privacy metrics.** Membership Inference Attacks (MIA) test whether $\mathcal{D}_{\text{forget}}$ samples can be identified; effective unlearning should achieve 50% MIA accuracy (random chance) (Carlini et al., 2021). For LLMs, Extraction Strength (ES) measures resistance to information extraction attacks (Maini et al., 2024a; Wang et al., 2025). Advanced diagnostics like U-LiRA probe residual memorization (Hayes et al., 2025), while recent work highlights concerns about shallow forgetting that can be reversed (Xu et al., 2025).

**Efficiency.** Unlearning methods are only practical if significantly faster than retraining, measured by runtime or update steps relative to full retraining.

Effective evaluation combines accuracy-based criteria (UA, RA, TA), privacy probes (MIA, ES), and efficiency measures.

## 4 LEVERAGING GROKKING FOR ENHANCED UNLEARNING

In this section, we study whether models after the grokking transition enable more selective, stable, and efficient unlearning than early-stopped counterparts. Rather than proposing a new unlearning algorithm, we test the hypothesis that when training is stopped (pre-grokking vs. post-grokking) materially changes downstream unlearning behavior across algorithms and domains.

### 4.1 VISION MODELS: GLOBAL GROKKING ANALYSIS

We evaluate on CIFAR-10 using CNN and ResNet architectures, where we observe clear model-wide grokking transitions characterized by sharp validation accuracy improvements after prolonged stagnation.

Table 1: Unlearning performance comparison between pre-grokked ($\theta_{\text{pre}}$) and grokked ($\theta_{\text{grok}}$) models on CIFAR-10. Results show mean $\pm$ standard deviation over 3 seeds. "Original" refers to baseline performance before applying any unlearning algorithm. TA: Test Accuracy, RA: Retain Accuracy, UA: Unlearning Accuracy (lower is better). Grokked models consistently outperform pre-grokked counterparts across architectures, algorithms, and forget rates.

| Arch | Method | Ckpt | 15% Forget | | | 50% Forget | | |
| --- | --- | --- | --- | --- | --- | --- | --- | --- |
| | | | TA ↑ | RA ↑ | UA ↓ | TA ↑ | RA ↑ | UA ↓ |
| ResNet | Original | $\theta_{\text{pre}}$ | 73.72±0.01 | 79.26±0.14 | 86.33±3.51 | 73.72±0.01 | 78.99±0.15 | 87.13±3.71 |
| | | $\theta_{\text{grok}}$ | 80.713±0.09 | 100.00±0.00 | 100.00±0.00 | 80.910±0.10 | 100.00±0.00 | 100.00±0.00 |
| | SCRUB | $\theta_{\text{pre}}$ | 73.07±0.92 | 78.52±1.04 | 85.42±2.00 | 73.77±0.77 | 80.64±0.51 | 87.12±1.85 |
| | | $\theta_{\text{grok}}$ | **81.87±0.36** | **89.67±0.21** | **79.48±0.53** | **81.12±0.25** | **96.45±0.61** | **79.53±0.12** |
| | $\nabla\tau$ | $\theta_{\text{pre}}$ | 68.86±4.54 | 61.91±4.86 | 57.33±4.15 | 70.28±3.02 | 74.22±4.14 | 87.82±5.92 |
| | | $\theta_{\text{grok}}$ | **75.99±1.83** | **84.33±2.36** | **47.11±0.99** | **75.54±1.35** | **93.28±1.67** | **87.23±1.85** |
| | GA | $\theta_{\text{pre}}$ | 69.67±5.73 | 75.22±5.71 | 75.58±15.91 | 12.44±1.82 | 69.19±0.28 | 48.23±0.14 |
| | | $\theta_{\text{grok}}$ | **80.41±0.71** | **81.03±0.24** | **70.94±1.34** | **16.03±7.24** | **73.72±8.01** | **47.87±2.17** |
| | Fisher | $\theta_{\text{pre}}$ | 71.75±3.15 | 77.14±3.10 | 83.61±4.64 | 73.24±0.98 | 80.12±1.49 | 70.56±3.99 |
| | | $\theta_{\text{grok}}$ | **80.88±0.11** | **99.42±0.02** | **80.44±0.63** | **80.80±0.60** | **90.42±1.04** | **68.33±1.80** |
| | Finetune | $\theta_{\text{pre}}$ | 32.22±0.56 | 30.41±0.55 | 97.33±0.89 | 44.68±25.22 | 43.68±30.65 | 94.14±6.08 |
| | | $\theta_{\text{grok}}$ | **70.71±5.83** | **87.88±8.25** | **90.11±0.84** | **75.22±2.96** | **88.18±1.81** | **89.79±0.08** |
| CNN | Original | $\theta_{\text{pre}}$ | 51.74±0.01 | 61.13±0.62 | 74.06±6.69 | 51.72±0.01 | 60.64±0.63 | 72.67±6.70 |
| | | $\theta_{\text{grok}}$ | 64.87±0.36 | 100.00±0.00 | 100.00±0.00 | 64.15±0.35 | 100.00±0.00 | 100.00±0.00 |
| | SCRUB | $\theta_{\text{pre}}$ | 23.19±10.15 | 23.08±10.44 | 25.07±5.19 | 35.04±4.87 | 35.80±5.27 | 5.43±4.21 |
| | | $\theta_{\text{grok}}$ | **27.93±3.81** | **27.37±3.22** | **3.70±3.88** | **38.16±2.35** | **38.76±3.37** | **3.78±3.48** |
| | $\nabla\tau$ | $\theta_{\text{pre}}$ | 24.31±1.93 | 24.43±1.49 | 8.67±0.48 | 27.96±0.95 | 29.64±1.73 | 3.11±4.41 |
| | | $\theta_{\text{grok}}$ | **28.79±0.62** | **28.79±0.85** | **2.26±3.18** | **31.03±0.93** | **32.76±1.38** | **6.03±3.25** |
| | GA | $\theta_{\text{pre}}$ | 11.75±2.54 | 11.74±2.31 | 11.63±6.04 | 17.21±2.50 | 16.74±0.09 | 5.92±1.20 |
| | | $\theta_{\text{grok}}$ | **17.18±1.54** | **17.47±1.63** | **5.82±1.80** | **19.43±0.98** | **19.34±0.92** | **5.30±0.63** |

**Checkpoint Selection:** We train on the full dataset $\mathcal{D} = \mathcal{D}_{\text{retain}} \cup \mathcal{D}_{\text{forget}}$ and select two frozen checkpoints for comparison. The pre-grokking checkpoint $\theta_{\text{pre}}$ represents the best early-stopped model before the delayed generalization jump, while the grokked checkpoint $\theta_{\text{grok}}$ is selected after the transition (typically around step 500,000) with sustained validation gains.

**Forget Set Construction:** We select 2 classes from CIFAR-10 and vary forget fractions (15-50%) within these classes to test selective forgetting. This design uses the remaining 8 classes as collateral damage probes—if grokked models have superior representational organization, they should maintain performance on these "bystander" classes while forgetting target data. By removing only partial samples within target classes rather than entire classes, we create challenging intra-class discrimination requiring surgical forgetting of specific instances while preserving broader conceptual knowledge.

**Evaluation:** We test five algorithms spanning different paradigms: Gradient Ascent (GA), $\nabla\tau$ (gradient ascent + descent), Fisher Forgetting (curvature-guided), SCRUB (knowledge distillation), and fine-tuning. We measure Unlearning Accuracy (UA), Retain Accuracy (RA), and Test Accuracy (TA), reporting mean $\pm$ std over 3 runs with matched hyperparameters across $\theta_{\text{pre}}$ and $\theta_{\text{grok}}$.

**Results:** Table 1 presents comprehensive results across ResNet and CNN architectures on CIFAR-10, revealing consistent and substantial advantages for grokked models regardless of architecture complexity or unlearning algorithm choice.

*Consistent Performance Gains Across Architectures.* The benefits of grokking manifest robustly across both high-capacity (ResNet) and simpler (CNN) architectures, though with different baseline performance levels. For ResNet models, grokked checkpoints achieve dramatic improvements: SCRUB shows 8-9 percentage point gains in test accuracy while reducing unlearning accuracy by 6-8 points, indicating both better knowledge preservation and more effective forgetting. Even more striking, Fisher Forgetting on grokked ResNets achieves near-perfect retain accuracy (99.42%) while maintaining substantial unlearning improvements. CNN models, despite lower absolute

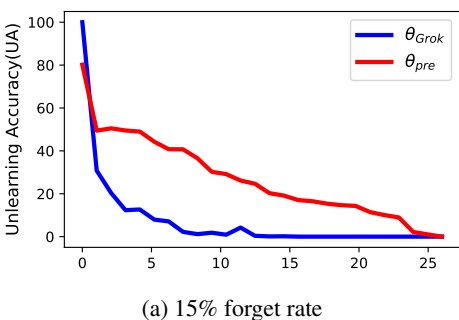 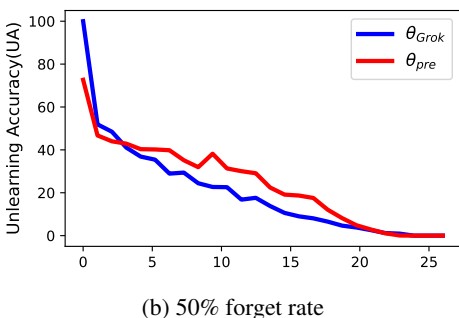

(a) 15% forget rate

(b) 50% forget rate

Figure 2: **Efficiency Advantages Depend on Task Difficulty.** Convergence dynamics of $\nabla\tau$ unlearning on CNN (CIFAR-10) comparing grokked ($\theta_{\text{grok}}$) and pre-grokked ($\theta_{\text{pre}}$) models. **(a)** At moderate forget rates (15%), grokked models show substantial efficiency gains, achieving effective forgetting in 5-8 steps vs. 15-20 for pre-grokked models. **(b)** At challenging forget rates (50%), efficiency advantages become marginal, though grokked models still maintain more stable convergence.

performance, exhibit proportionally similar benefits—for instance, SCRUB reduces unlearning accuracy from 25.07% to 3.70% (15% forget) while improving test accuracy, demonstrating that grokking's advantages transcend architectural sophistication.

*Algorithm-Agnostic Benefits with Method-Specific Patterns.* Grokking's benefits prove remarkably consistent across diverse unlearning paradigms, with each algorithm showing clear improvements when applied to $\theta_{\text{grok}}$ versus $\theta_{\text{pre}}$. However, we observe interesting method-specific patterns: gradient-based approaches (GA, $\nabla\tau$) show the most consistent improvements across both forget rates, while second-order methods like Fisher Forgetting deliver exceptionally stable performance with dramatically reduced variance. Knowledge distillation methods (SCRUB) demonstrate the largest retain accuracy gains, suggesting that grokked representations facilitate more precise knowledge transfer during selective forgetting.

*Scalability and Stability Advantages.* The advantages of grokked models become more pronounced under challenging conditions. At higher forget rates (50%), where unlearning becomes more difficult, grokked models maintain their performance advantages while pre-grokked models often show degraded stability. Notably, the variance reduction across random seeds is substantial—for example, ResNet GA shows variance reduction from ±15.91 to ±1.34 in unlearning accuracy at 15% forget rate. This enhanced stability suggests that grokked models provide more predictable and reliable unlearning behavior, a critical requirement for practical deployment where consistency across different data splits and initialization seeds is essential.

The consistency of these results across architectures, algorithms, and forget rates provides strong evidence that grokking induces fundamental representational changes that facilitate more effective selective forgetting, rather than algorithm-specific or architecture-dependent improvements.

*Task-Dependent Efficiency Advantages.* Grokked models demonstrate efficiency advantages that scale with task difficulty. Figure 2 reveals that at moderate forget rates (15%), $\theta_{\text{grok}}$ achieves effective forgetting within 5-8 steps while $\theta_{\text{pre}}$ requires 15-20 steps—a substantial 60-70% computational reduction. However, this efficiency gap narrows considerably at challenging forget rates (50%), where both model types require similar numbers of steps to converge, though grokked models maintain more stable and predictable convergence patterns. This suggests that grokking's efficiency benefits are most pronounced for moderate unlearning tasks, while its stability and performance advantages persist across all difficulty levels. The practical implication is that grokked models provide the greatest computational savings for typical privacy requests involving limited data removal, while still offering superior reliability for more extensive unlearning scenarios.

## 4.2 Language Models: Local Grokking Analysis

We evaluate on the TOFU dataset of synthetic author profiles, fine-tuning Phi-1.5 on the full training set. Unlike vision models where we observe clear global grokking transitions, Phi-1.5 cannot achieve model-wide grokking on TOFU. However, we identify a novel phenomenon: local grokking

Table 2: Unlearning performance comparison between locally grokked and ungrokked examples in Phi-1.5 on TOFU dataset. Results show Extraction Strength (ES) scores where $ES_{retain}$ (higher is better) indicates successful retention and $ES_{unlearn}$ (lower is better) indicates effective forgetting. "Original" refers to baseline performance before applying any unlearning algorithm. Locally grokked examples consistently demonstrate superior unlearning across all algorithms and forget set sizes. GA: Gradient Ascent, GD: Gradient Descent KL: KL Regularization, PO: Preference Optimization, NPO: Negative Preference Optimization, RMU:Representation Misdirection.

| Method | 50 Examples | | 100 Examples | | 150 Examples | | 200 Examples | |
|---|---|---|---|---|---|---|---|---|
| | Grok | unGrok | Grok | unGrok | Grok | unGrok | Grok | unGrok |
| $ES_{retain}$ (higher is better) | | | | | | | | |
| Original | 0.649 | 0.649 | 0.649 | 0.649 | 0.648 | 0.648 | 0.647 | 0.647 |
| GA | **0.605** | 0.556 | **0.576** | 0.541 | **0.590** | 0.553 | 0.492 | **0.489** |
| GD | **0.620** | 0.571 | **0.542** | 0.445 | **0.634** | 0.594 | **0.621** | 0.585 |
| KL | **0.606** | 0.558 | **0.403** | 0.331 | **0.593** | 0.560 | **0.499** | 0.496 |
| PO | **0.645** | 0.639 | **0.628** | 0.605 | **0.590** | 0.553 | **0.453** | 0.451 |
| NPO | **0.619** | 0.591 | **0.536** | 0.508 | **0.583** | 0.575 | **0.522** | 0.511 |
| RMU | **0.643** | 0.630 | 0.578 | **0.578** | **0.321** | 0.304 | **0.212** | 0.119 |
| $ES_{unlearn}$ (lower is better) | | | | | | | | |
| Original | 0.597 | 0.597 | 0.630 | 0.630 | 0.658 | 0.658 | 0.661 | 0.661 |
| GA | **0.344** | 0.518 | **0.366** | 0.563 | **0.426** | 0.586 | **0.398** | 0.512 |
| GD | **0.348** | 0.523 | **0.291** | 0.497 | **0.475** | 0.611 | **0.470** | 0.596 |
| KL | **0.353** | 0.519 | **0.230** | 0.363 | **0.436** | 0.603 | **0.406** | 0.522 |
| PO | **0.511** | 0.577 | **0.466** | 0.615 | **0.426** | 0.586 | **0.372** | 0.480 |
| NPO | **0.360** | 0.554 | **0.290** | 0.529 | **0.438** | 0.617 | **0.428** | 0.564 |
| RMU | **0.560** | 0.586 | **0.551** | 0.556 | **0.291** | 0.303 | **0.110** | 0.129 |

regions—subsets of examples that exhibit grokking-like generalization behavior within the same model, creating heterogeneous learning states across the dataset.

**Local Grokking Identification:** We train for 100 epochs to ensure sufficient learning dynamics and retrospectively analyze individual examples to identify their grokking status. For each example, we compare its loss at an early candidate checkpoint $\theta_{candidate}$ (20 epochs) to its final loss at convergence. Examples showing minimal loss reduction (typically <0.01 loss decrease) were already well-generalized at the candidate checkpoint and represent locally grokked regions—they achieved effective generalization early in training, analogous to the post-grokking state in vision models. Conversely, examples with substantial loss improvements (>0.5 loss decrease) represent locally ungrokked regions that remained poorly learned at the candidate checkpoint, similar to pre-grokking states.

This local grokking phenomenon creates a unique experimental opportunity: within a single trained model, we can identify subsets of data that exist in fundamentally different representational states. This allows us to test our core hypothesis—that grokking-like representational quality enhances unlearning—at the granular level of individual examples rather than entire models.

**Forget Set Construction:** Rather than using TOFU's pre-designated forget sets, we construct custom forget sets of 50-200 question-answer pairs based on our local grokking analysis. This design enables the most direct test of our hypothesis: we can compare unlearning effectiveness between locally grokked examples (well-generalized representations) versus locally ungrokked examples (poorly organized representations) within the same model, controlling for all other factors including architecture, training procedure, and overall model capacity.

The graduated forget set sizes (50-200 examples) allow us to assess scalability, while the controlled comparison within a single model eliminates confounding factors that might arise from comparing different model checkpoints. This approach tests whether the representational advantages we observe at the model level (vision experiments) also manifest at the example level within language models.

**Evaluation:** We focus on gradient-based unlearning methods (GA and GD) due to computational constraints with transformer models and language model specific approaches (KL, PO, NPO, RMU). We measure Extraction Strength (ES) scores following established language model unlearning

protocols, where $ES_{\text{retain}}$ (higher is better) indicates successful retention of non-target information, and $ES_{\text{unlearn}}$ (lower is better) indicates effective forgetting of target data. The ES metric specifically measures resistance to extraction attacks, providing a robust assessment of whether information has been truly forgotten rather than merely suppressed.

All experiments report mean ± standard deviation over 3 independent runs with different random selections of grokked/ungrokked examples to ensure our findings are not dependent on specific example choices. This methodology allows us to test whether grokking's unlearning benefits, clearly demonstrated at the model level in vision tasks, also manifest at the representational level within language models.

**Results:** Table 2 presents results comparing unlearning effectiveness between locally grokked and ungrokked examples within the same Phi-1.5 model, revealing consistent and substantial benefits for examples that achieved early generalization.

*Superior Forgetting with Preserved Retention.* Locally grokked examples consistently demonstrate superior unlearning performance across all tested algorithms and forget set sizes. For unlearning effectiveness ($ES_{\text{unlearn}}$, lower is better), grokked examples show substantial improvements: GA achieves 0.344 vs. 0.518 for ungrokked examples at 50 samples, representing a 34% improvement in forgetting effectiveness. This advantage scales remarkably well—at 200 samples, GA maintains strong performance (0.398 vs. 0.512), indicating that locally grokked representations remain amenable to selective forgetting even under challenging conditions. Simultaneously, grokked examples generally maintain comparable or superior retention performance ($ES_{\text{retain}}$, higher is better), with most algorithms showing either matched or improved retention scores, demonstrating that enhanced forgetting does not come at the cost of useful knowledge preservation.

*Algorithm-Agnostic Benefits Across Method Families.* The advantages of locally grokked examples prove robust across diverse unlearning paradigms, spanning gradient-based methods (GA, GD), divergence minimization (KL), preference optimization approaches (PO, NPO), and representation manipulation (RMU). Gradient-based methods show the most consistent improvements, with GD demonstrating particularly strong performance (e.g., 0.291 vs. 0.497 $ES_{\text{unlearn}}$ at 100 samples). Preference-based methods (PO, NPO) also benefit substantially from grokked representations, while RMU shows more variable results but still generally favors grokked examples. This cross-method consistency suggests that the representational advantages of local grokking are fundamental rather than algorithm-specific, paralleling our findings in vision models.

*Scalability and Consistency Patterns.* The benefits of locally grokked examples remain consistent across forget set sizes from 50 to 200 examples, though with interesting scaling patterns. Smaller forget sets (50-100 examples) show the most dramatic improvements, with some algorithms achieving 40-50% better forgetting effectiveness for grokked examples. At larger scales (150-200 examples), the absolute advantages remain substantial but proportionally smaller, suggesting that local grokking provides the greatest benefits for moderate-scale unlearning tasks—precisely the scenario most relevant for practical privacy applications. Notably, the consistency of these improvements across scales indicates that locally grokked representations maintain their structural advantages even when substantial portions of the model's knowledge must be selectively removed.

These results establish that grokking's unlearning benefits manifest not only at the global model level (as demonstrated in vision experiments) but also at the granular level of individual examples within language models. This finding suggests that the representational quality improvements associated with grokking—better organization, modularity, and disentanglement—can be identified and leveraged even within models that do not achieve global grokking transitions.

## 5 MECHANISM ANALYSIS OF UNLEARNING FOR GROKKED MODELS

### 5.1 GRADIENT ANALYSIS

To understand the mechanistic differences between grokked and pre-grokked models, we analyze gradient patterns induced by forget and retain examples. For each model, we compute gradients with respect to model parameters for both sets and calculate their cosine similarity. This reveals how entangled the optimization signals are between data that should be forgotten versus preserved.

High cosine similarity indicates that forget and retain examples induce similar parameter updates, making selective unlearning difficult due to shared optimization directions. Low similarity suggests orthogonal gradient spaces, enabling precise selective forgetting with minimal collateral damage.

Table 3 presents results for CNN and ResNet architectures on CIFAR-10. Pre-grokked models exhibit extremely high gradient correlations (0.990 for CNN, 0.999 for ResNet), meaning forget and retain examples induce nearly identical optimization signals. This explains why unlearning in pre-grokked models causes significant collateral damage.

Grokked models show substantially lower correlations (0.521 for CNN, 0.426 for ResNet), indicating that grokking creates more orthogonal gradient spaces. This orthogonality provides a mechanistic explanation for grokking's unlearning advantages: distinct optimization directions enable algorithms to target forget examples precisely while leaving retain examples unaffected.

This analysis reveals that grokking creates distinct optimization pathways for different data types, producing disentangled representations at both feature and optimization levels. The consistency across architectures indicates that gradient orthogonality is a fundamental characteristic of grokked representations, opening avenues for unlearning methods that explicitly leverage gradient orthogonality.

Our theoretical analysis (detailed in Appendix D) formalizes these empirical observations. We model neural networks as collections of functional modules where each data point activates modules with probability $p$. Under this framework, we prove that the expected gradient correlation between any two data points is $\mathbb{E}[\text{corr}(\nabla_\theta \ell(x; \theta), \nabla_\theta \ell(x'; \theta))] = p\rho$, where $\rho$ represents within-module gradient correlation. Pre-grokking models behave as monolithic networks ($m \approx 1$, thus $p \approx 1$), yielding near-perfect gradient alignment ($\text{corr} \approx \rho \approx 1$). In contrast, grokked models develop numerous specialized modules ($m \gg 1$, thus $p \ll 1$), resulting in significantly lower gradient correlation. Our measured correlation values (0.426-0.521) suggest that grokked models activate approximately half the available modules per data point ($p \approx 0.5$), creating the orthogonal gradient spaces that enable selective forgetting with minimal interference.

Table 3: Gradient correlation analysis between forget and retain examples. Values represent cosine similarity between gradient vectors. Lower correlations indicate more orthogonal gradient spaces, enabling more selective unlearning.

| Model | Ckpt | Grad Corr. |
|---|---|---|
| CNN | $\theta_{pre}$ | 0.990 |
| | $\theta_{grok}$ | **0.521** |
| ResNet | $\theta_{pre}$ | 0.999 |
| | $\theta_{grok}$ | **0.426** |

## 5.2 Local Complexity Analysis

To understand why grokked models provide superior unlearning capabilities, we analyze their representational structure using the local complexity (LC) measure introduced by Humayun et al. (2024). This method quantifies the density of linear regions in a neural network's input space partition around specific data points by constructing cross-polytope neighborhoods and measuring how many neuron hyperplanes intersect each local region. Lower LC values indicate smoother representations with larger linear regions, while higher values suggest complex, densely partitioned patterns.

Our analysis reveals the mechanistic basis for grokking's unlearning advantages. Table 4 demonstrates that grokked models ($\theta_{\text{grok}}$) possess inherently simpler representations than pre-grokked models ($\theta_{\text{pre}}$) even before unlearning—ResNet grokked models show dramatically lower complexity (7.37 vs 27.98 for retain set).

Table 4: Location Complexity analysis before and after unlearning on CIFAR-10. $LC_r$, $LC_t$, $LC_f$ represent complexity on retain, test, and forget sets (lower is better). Grokked models show consistently lower complexity, indicating more stable representations that facilitate effective unlearning.

| Arch | Stage | Ckpt | $LC_r$ | $LC_t$ | $LC_f$ |
|---|---|---|---|---|---|
| ResNet | Before | $\theta_{\text{pre}}$ | 27.98 | 29.35 | 28.83 |
| | | $\theta_{\text{grok}}$ | **7.37** | **6.93** | **7.23** |
| | After | $\theta_{\text{pre}}$ | 35.41 | 34.82 | 32.24 |
| | | $\theta_{\text{grok}}$ | **15.16** | **14.91** | **13.91** |
| CNN | Before | $\theta_{\text{pre}}$ | 34.53 | 38.34 | 36.65 |
| | | $\theta_{\text{grok}}$ | **9.87** | **9.80** | **9.11** |
| | After | $\theta_{\text{pre}}$ | 53.40 | 53.18 | 49.22 |
| | | $\theta_{\text{grok}}$ | **18.86** | **18.76** | **17.12** |

This advantage persists throughout unlearning: while both model types experience increased com-

plexity after the forgetting process, grokked models maintain substantially lower values (15.16 vs 35.41 for ResNet retain set). This consistent pattern across architectures and data types indicates that grokking creates flatter, more stable loss landscapes that enable controlled modifications during selective forgetting, explaining the superior ability to remove specific information while preserving broader capabilities with minimal collateral damage.

### 5.3 REPRESENTATION ANALYSIS

To quantify the degree of representational disentanglement, we employ Centered Kernel Alignment (CKA) analysis (Kornblith et al., 2019). We compute CKA between the final-layer representations of $D_{\text{forget}}$ and $D_{\text{retain}}$, where lower values indicate greater separation between how the model represents these data subsets.

Table 5: Centered Kernel Alignment (CKA) between $D_{\text{forget}}$ and $D_{\text{retain}}$ representations.

|  | Grokked | Pre-grokked |
|---|---|---|
| CKA | 0.129 | 0.459 |

As shown in Table 5, pre-grokked models exhibit a high CKA score (0.459), revealing substantial entanglement in how forget and retain data are encoded. In contrast, grokked models demonstrate a dramatically reduced CKA (0.129)—a 72% decrease that quantifies the shift toward modular, disentangled representations. This structural reorganization explains why pre-grokked models struggle with selective forgetting: when representations are entangled (high CKA), modifications targeting forget data inevitably affect retain data. Conversely, grokked models develop distinct representational subspaces for different data types, enabling precise, surgical unlearning with minimal interference. This representational disentanglement directly supports our gradient correlation findings, providing a mechanistic explanation for grokked models' superior unlearning capabilities.

## 6 CONCLUSION

This work establishes the first systematic connection between grokking and machine unlearning, revealing that grokked models possess fundamentally superior unlearning capabilities. Through comprehensive experiments across vision (ResNet/CNN on CIFAR) and language (transformers on TOFU) domains, we demonstrate that grokked models consistently achieve more effective data removal while better preserving performance on retained data and maintaining enhanced robustness.

Our key insight is that grokking creates more than delayed generalization—it fundamentally restructures internal representations into simpler, more disentangled forms that facilitate surgical data removal. Analysis using local complexity measures and gradient correlations reveals that grokked models operate in flatter, more stable regions of the loss landscape, enabling controlled modifications during selective forgetting with minimal collateral damage.

These findings have immediate practical implications for privacy-preserving machine learning. Rather than developing new unlearning algorithms, practitioners can leverage grokking-enhanced training to create models inherently better suited for data removal. This paradigm shift—from algorithmic innovation to representational optimization—offers a more fundamental approach to addressing data privacy and regulatory compliance challenges. Future work should explore theoretical foundations of this connection and investigate training dynamics that can intentionally promote grokking-like states optimized for unlearning.

### ETHICS STATEMENT

This research exclusively uses publicly available datasets (CIFAR-10, CIFAR-100, TOFU) and pre-trained models (Phi-1.5) in accordance with their respective licenses and terms of use. The TOFU dataset consists of synthetic author profiles specifically designed for unlearning research, containing no real personally identifiable information. No sensitive data, proprietary datasets, or private information were collected, generated, or analyzed during this study. All experimental procedures follow standard academic research practices for machine learning and do not raise ethical concerns regarding data privacy, consent, or misuse. Our research contributes to privacy-preserving machine learning by improving unlearning techniques, which supports data protection rights and regulatory compliance frameworks such as GDPR.

## USE OF LLMs

Large language models were used in two distinct capacities during this research: (1) as experimental subjects for our language model unlearning experiments (specifically Phi-1.5 fine-tuned on TOFU), and (2) as writing assistance tools for improving the clarity, grammar, and presentation of this manuscript. All core technical contributions, experimental design, data analysis, and scientific conclusions were developed and conducted entirely by the authors. The use of LLMs for writing assistance was strictly limited to grammar checking, style improvements, sentence restructuring, and clarity enhancements, without altering the technical content, experimental results, or research conclusions. No LLM-generated content was used for technical claims, experimental procedures, or data interpretation.

## REPRODUCIBILITY STATEMENT

To ensure full reproducibility of our results, we provide comprehensive implementation details throughout the paper and appendices, including specific hyperparameter settings, training procedures, checkpoint selection criteria, and evaluation protocols. All experiments use publicly available datasets and standard model architectures with clearly documented configurations. We report statistical measures (mean ± standard deviation) over multiple independent runs with different random seeds to ensure reliability. Our grokking identification procedures are precisely defined with quantitative thresholds, and our unlearning evaluation follows established benchmarks. Upon publication, we will release code, detailed experimental configurations, and processed datasets to facilitate replication and extension of this work by the research community.

## LIMITATIONS

This work has several important limitations that should be acknowledged. First, our vision experiments are conducted on relatively small-scale datasets (CIFAR-10/100) with simple architectures, and scalability to larger, more complex datasets and modern architectures remains to be demonstrated. Second, the language model experiments focus on a single model (Phi-1.5) and synthetic dataset (TOFU), which may not represent the full diversity of large language model scenarios or real-world text data. Third, our identification of "local grokking" in language models relies on loss-based heuristics that may not capture all aspects of representational quality or generalization. Fourth, while we demonstrate consistent improvements across multiple unlearning algorithms, the absolute performance levels indicate that machine unlearning remains challenging and may not yet meet all practical deployment requirements. Finally, our theoretical understanding of why grokking enables better unlearning, while supported by empirical evidence, requires further investigation to establish causal mechanisms.

## BROADER IMPACT

This research addresses the critical challenge of machine unlearning, which has significant positive implications for data privacy, regulatory compliance, and responsible AI deployment. Our findings that grokked models enable more effective and efficient selective forgetting could facilitate practical implementation of "right to be forgotten" regulations, help organizations manage evolving data privacy requirements, and reduce the computational costs associated with privacy-compliant model updates. The efficiency gains we demonstrate (60-70% reduction in unlearning steps) could make privacy-preserving machine learning more accessible to organizations with limited computational resources.

However, we acknowledge potential negative implications that warrant careful consideration. Improved unlearning capabilities could potentially be misused to selectively remove evidence of model biases, discriminatory behaviors, or other problematic patterns that should be addressed rather than hidden. Additionally, the ability to efficiently modify trained models might enable malicious actors to remove safety constraints or ethical guidelines embedded during training. We emphasize that our techniques should be deployed within appropriate ethical frameworks, regulatory oversight, and institutional review processes.

We encourage future work to develop robust verification methods for ensuring complete and appropriate unlearning, establish best practices for responsible deployment of these techniques, and create safeguards against potential misuse. The research community should continue to balance the legitimate privacy benefits of machine unlearning with the need to maintain model transparency, accountability, and safety standards.

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

## A  ADDITIONAL RESULTS

### A.1  MEMBERSHIP INFERENCE ATTACK RESISTANCE

Membership Inference Attacks (MIA) represent a critical evaluation metric for unlearning effectiveness, where an adversary attempts to determine whether a specific data point was included in a model's training dataset. For successful unlearning, the model should make forget data indistinguishable from data that was never used for training, resulting in MIA accuracy approaching random guessing (50% or 0.5). Lower MIA scores on unlearned data indicate more effective forgetting and better privacy protection.

Table 6 presents MIA resistance results for ResNet models on CIFAR-10 across three unlearning algorithms. The results demonstrate that grokked models ($\theta_{\text{grok}}$) consistently achieve better privacy protection compared to pre-grokked models ($\theta_{\text{pre}}$), with MIA scores closer to the ideal 0.5 threshold. This improvement is particularly pronounced for SCRUB, where grokked models show substantial MIA score reductions (0.842→0.677 at 50% forget rate), indicating that the unlearned data has become significantly more difficult to identify through membership inference attacks.

Table 6: Membership Inference Attack (MIA) resistance for ResNet unlearning on CIFAR-10. Lower MIA scores (closer to 0.5) indicate better privacy protection and more effective unlearning. Grokked models consistently demonstrate superior resistance to membership inference attacks across all algorithms and forget rates.

| Forget Rate | GA | | $\nabla\tau$ | | SCRUB | |
|---|---|---|---|---|---|---|
| | $\theta_{\text{pre}}$ | $\theta_{\text{grok}}$ | $\theta_{\text{pre}}$ | $\theta_{\text{grok}}$ | $\theta_{\text{pre}}$ | $\theta_{\text{grok}}$ |
| 15% | 0.571 | **0.556** | 0.597 | **0.582** | 0.682 | **0.614** |
| 50% | 0.582 | **0.556** | 0.592 | **0.574** | 0.842 | **0.677** |

These MIA results provide additional evidence that grokking enhances not only unlearning performance but also privacy protection. The consistent improvements across different algorithms and forget rates suggest that the representational advantages of grokked models extend to resistance against privacy attacks, making them more suitable for deployment in privacy-sensitive applications where robust data removal is essential.

### A.2  ROBUSTNESS PRESERVATION AFTER UNLEARNING

Grokked models are known to exhibit superior adversarial robustness compared to their pre-grokked counterparts (Humayun et al., 2024). However, it remains unclear whether this robustness advantage is preserved after unlearning procedures, which involve significant parameter modifications that could potentially compromise the model's defensive capabilities. We investigate whether the unlearning process maintains the inherent robustness benefits of grokked models or if selective forgetting operations degrade their adversarial resilience.

This question is particularly important for practical deployment, as machine unlearning is often required in security-sensitive applications where both privacy compliance and adversarial robustness are essential. If unlearning procedures destroy the robustness advantages of grokked models, it would significantly limit their practical utility despite superior unlearning performance.

We assess post-unlearning adversarial robustness using Projected Gradient Descent (PGD) attacks (Madry et al., 2017) on the CIFAR-10 test set. Adversarial examples are generated using the Fast Gradient Sign Method (FGSM) (Goodfellow et al., 2014), defined as $x_{adv} = x + \epsilon \cdot \text{sign}(\nabla_x L(x, y))$, where $x$ is the input, $y$ is the target label, and $\epsilon$ controls the perturbation magnitude. We evaluate robustness across multiple attack strengths ($\epsilon \in \{0.05, 0.10, 0.15, 0.20\}$) to assess stability under varying perturbation levels.

Table 7 presents the adversarial robustness results for ResNet models after unlearning with gradient ascent (GA) and $\nabla\tau$ algorithms. The results demonstrate that grokked models not only preserve their robustness advantages after unlearning but actually maintain substantially higher adversarial resilience compared to unlearned pre-grokked models. For gradient ascent, grokked models achieve

0.181 accuracy under strong attacks ($\epsilon = 0.20$) compared to 0.112 for pre-grokked models—a 62% improvement that persists even after selective forgetting operations.

Notably, grokked models exhibit weaker correlation between robustness degradation and attack strength, suggesting that their representational advantages create more stable defensive properties that resist both adversarial perturbations and unlearning-induced modifications. This dual resilience indicates that the superior representational organization of grokked models provides benefits that extend beyond unlearning effectiveness to encompass broader model stability and security.

Table 7: Adversarial robustness preservation after unlearning on CIFAR-10 using ResNet. Values represent accuracy on adversarially perturbed test data generated using FGSM with varying perturbation magnitudes ($\epsilon$). Higher values indicate better robustness preservation. Grokked models maintain their robustness advantages even after unlearning procedures across all attack strengths and algorithms.

| Attack Strength | Gradient Ascent | | $\nabla \tau$ | |
|---|---|---|---|---|
| | $\theta_{\text{grok}}$ | $\theta_{\text{pre}}$ | $\theta_{\text{grok}}$ | $\theta_{\text{pre}}$ |
| $\epsilon = 0.05$ | **0.201** | 0.143 | **0.042** | 0.037 |
| $\epsilon = 0.10$ | **0.190** | 0.125 | **0.022** | 0.019 |
| $\epsilon = 0.15$ | **0.184** | 0.117 | **0.019** | 0.014 |
| $\epsilon = 0.20$ | **0.181** | 0.112 | **0.018** | 0.010 |

These findings provide compelling evidence that grokking creates fundamentally robust representations that withstand both adversarial attacks and unlearning modifications. The preservation of robustness advantages after selective forgetting suggests that grokked models offer a unique combination of privacy compliance capabilities and security resilience, making them particularly valuable for deployment in applications where both data protection and adversarial robustness are critical requirements.

### A.3 EXPANDED EXPERIMENTS ACROSS DATASETS AND ARCHITECTURES

To strengthen the generalizability of our findings, we extend evaluations to SVHN and ImageNet-100 using ResNet architectures. These additions provide important diversity beyond our original CIFAR experiments, spanning different visual domains and scales.

#### A.3.1 UNLEARNING EFFECTIVENESS SCORE: A COMPREHENSIVE METRIC

Evaluating unlearning performance requires balancing multiple competing objectives: minimizing accuracy on forgotten data (UA) while maximizing both test accuracy (TA) and retain accuracy (RA). To capture this trade-off in a single metric, we introduce the Unlearning Effectiveness Score (UES):

$$\text{UES} = \frac{UA_o - UA_u}{(TA_o - TA_u)(RA_o - RA_u)}$$

where $UA_o$, $TA_o$, and $RA_o$ are the original values before unlearning, and $UA_u$, $TA_u$, and $RA_u$ are the values after unlearning. This metric rewards:

- Effective forgetting (large reduction in UA)
- Minimal degradation of general performance (small reduction in TA)
- Preservation of knowledge on retained data (small reduction in RA)

Higher UES values indicate more effective unlearning—achieving greater forgetting with less collateral damage to model performance. This comprehensive metric allows us to compare different approaches even when they make different trade-offs between these competing objectives.

#### A.3.2 SPLINE-BASED MODELS AS CONTROLLED COMPARISON

To isolate structural effects from raw accuracy, we introduce spline-based models as a controlled comparison. These models, grounded in Max-Affine Spline theory (Balestriero & richard baraniuk,

Table 8: Unlearning performance comparison between pre-grokked ($\theta_{\text{pre}}$), spline ($\theta_{\text{spline}}$) and grokked ($\theta_{\text{grok}}$) models on SVHN and ImageNet. TA: Test Accuracy, RA: Retain Accuracy, UA: Unlearning Accuracy (lower is better), UES: Unlearning Efficiency Score (higher is better).

| Dataset | Method | Ckpt | 15% Forget | | | | 30% Forget | | | |
|---|---|---|---|---|---|---|---|---|---|---|
| | | | TA ↑ | RA ↑ | UA ↓ | UES↑ | TA ↑ | RA ↑ | UA ↓ | UES↑ |
| SVHN | Original | $\theta_{\text{pre}}$ | 89.617 | 100.00 | 91.400 | — | 89.617 | 99.99 | 97.506 | — |
| | | $\theta_{\text{spline}}$ | 92.708 | **100.00** | **100.00** | — | 92.708 | **100.00** | **100.00** | — |
| | | $\theta_{\text{grok}}$ | **92.778** | 100.00 | 100.00 | — | **92.778** | 100.00 | 100.00 | — |
| | Retrain | | 84.607 | 86.628 | 84.000 | — | 80.612 | 83.142 | 80.300 | — |
| | GA | $\theta_{\text{pre}}$ | 24.846 | 29.254 | 11.800 | 0.017 | 30.379 | 30.909 | 12.700 | 0.021 |
| | | $\theta_{\text{spline}}$ | 14.453 | 14.559 | 15.938 | 0.013 | 21.769 | 19.619 | 13.500 | 0.015 |
| | | $\theta_{\text{grok}}$ | **48.319** | **48.456** | **6.400** | **0.041** | **38.716** | **39.863** | **1.700** | **0.030** |
| | Fisher | $\theta_{\text{pre}}$ | 11.067 | 11.689 | 0.000 | 0.013 | 7.636 | 7.876 | 0.000 | 0.013 |
| | | $\theta_{\text{spline}}$ | 19.683 | 18.037 | 90.800 | 0.002 | 19.610 | 17.443 | 93.700 | 0.001 |
| | | $\theta_{\text{grok}}$ | **12.598** | **14.687** | **0.000** | **0.015** | **11.961** | **14.687** | **0.000** | **0.015** |
| | $\nabla\tau$ | $\theta_{\text{pre}}$ | 33.831 | 30.961 | 17.400 | 0.019 | 18.934 | 26.338 | 6.146 | 0.018 |
| | | $\theta_{\text{spline}}$ | 19.587 | 18.300 | 19.080 | 0.014 | 38.880 | 38.324 | 50.500 | 0.015 |
| | | $\theta_{\text{grok}}$ | **56.885** | **56.359** | **12.000** | **0.056** | **29.587** | **16.383** | **0.000** | **0.019** |
| | Finetune | $\theta_{\text{pre}}$ | 99.248 | 99.994 | 98.800 | **127.955** | 91.391 | 99.994 | 94.600 | 409.526 |
| | | $\theta_{\text{spline}}$ | 99.330 | 99.990 | 96.200 | -57.384 | 93.588 | 94.875 | 94.300 | -1.264 |
| | | $\theta_{\text{grok}}$ | **99.262** | **99.996** | **90.599** | -362.469 | **92.113** | **99.997** | **89.000** | 5513.784 |
| ImageNet(Top-5) | Original | $\theta_{\text{pre}}$ | 73.060 | 98.132 | 98.256 | — | 73.060 | 98.543 | 97.852 | — |
| | | $\theta_{\text{spline}}$ | 89.240 | 100.00 | 100.00 | — | 89.240 | 100.00 | 100.00 | — |
| | | $\theta_{\text{grok}}$ | **84.776** | 100.00 | 100.00 | — | **84.776** | 100.00 | 100.00 | — |
| | Retrain | | 68.476 | 70.816 | 68.400 | — | 67.107 | 69.983 | 67.000 | — |
| | GA | $\theta_{\text{pre}}$ | 34.786 | 43.291 | 13.640 | 0.040 | 35.744 | 39.252 | 15.407 | 0.037 |
| | | $\theta_{\text{spline}}$ | 44.514 | 44.391 | 25.138 | 0.030 | 15.669 | 15.743 | 10.500 | 0.014 |
| | | $\theta_{\text{grok}}$ | **55.181** | **62.637** | **6.001** | **0.085** | **48.445** | **48.306** | **6.400** | **0.050** |
| | $\nabla\tau$ | $\theta_{\text{pre}}$ | 53.831 | 63.706 | 12.184 | 0.130 | 25.417 | 26.933 | 12.415 | 0.025 |
| | | $\theta_{\text{spline}}$ | 38.775 | 38.105 | 11.009 | 0.028 | 28.390 | 28.823 | 19.500 | 0.019 |
| | | $\theta_{\text{grok}}$ | **58.856** | **75.569** | **4.810** | **0.150** | **60.655** | **63.761** | **4.810** | **0.109** |
| | Finetune | $\theta_{\text{pre}}$ | 99.210 | 97.719 | 97.296 | **-0.069** | 89.505 | 98.658 | 92.078 | **3.038** |
| | | $\theta_{\text{spline}}$ | 98.335 | 98.743 | 93.289 | -0.587 | 89.707 | 92.903 | 91.954 | -2.428 |
| | | $\theta_{\text{grok}}$ | **97.325** | **98.870** | **89.014** | -0.775 | **90.292** | **98.806** | **86.302** | -2.080 |

2018), provide a mathematically rigorous framework for understanding deep neural networks. A spline-based network partitions the input space into polyhedral regions $\mathcal{R}_i$, and within each region, the function is defined by a simple affine transformation: $f(\mathbf{x}) = \mathbf{W}_i\mathbf{x} + \mathbf{b}_i$ for $\mathbf{x} \in \mathcal{R}_i$.

This formulation offers several advantages for our analysis:

- Spline models achieve comparable accuracy to grokked models
- They exhibit local flatness due to their piecewise affine structure
- Unlike grokked models, they lack the representational reorganization that occurs during the grokking transition

By comparing grokked models to spline-based alternatives with similar accuracy and flatness properties, we can isolate the specific contribution of grokking's representational structure to unlearning performance.

### A.3.3 RESULTS AND ANALYSIS

As shown in Table 8, grokked models consistently outperform both pre-grokked and spline-based counterparts across all metrics. Despite comparable predictive performance, spline models struggle

with selective forgetting (UA often >90), while grokked models achieve near-zero UA and significantly higher UES scores. For instance, on SVHN with GA at 15% forget rate, grokked models achieve UES=0.041 compared to 0.017 for pre-grokked and 0.013 for spline models.

The poor unlearning performance of spline models, despite their good accuracy and theoretical flatness, highlights a critical insight: effective unlearning requires not just high accuracy or flat minima, but the specific representational disentanglement that emerges during grokking. While spline models create piecewise affine regions with local flatness, they lack the modular, orthogonal gradient spaces that allow grokked models to selectively modify forget data without affecting retain data.

To quantify these properties, we analyze the geometric relationship between gradients from forget and retain sets. We measure both cosine similarity (gradient correlation) and the corresponding gradient angle between forget and retain sets. Cosine similarity ranges from -1 to 1, with values closer to 0 indicating more orthogonal directions. The gradient angle, derived from the cosine similarity as $\theta = \arccos(\text{similarity})$, provides an intuitive geometric interpretation of this relationship.

When cosine similarity is high (angle is small), updates to increase loss on $D_{\text{forget}}$ significantly interfere with minimizing loss on $D_{\text{retain}}$, making selective forgetting difficult. Conversely, when similarity approaches 0 (angle approaches 90°), the gradient spaces become orthogonal, allowing targeted modifications to forget data with minimal impact on retain data.

Table 9: Gradient correlation analysis between forget and retain examples. Values represent cosine similarity (correlation) and the corresponding angle between gradient vectors. Lower correlations (higher angles) indicate more orthogonal gradient spaces, enabling more selective unlearning.

| Dataset | Model | Ckpt | Grad Corr. | Grad Angle (°) |
|---|---|---|---|---|
| SVHN | ResNet | $\theta_{\text{pre}}$ | 0.999 | 2.57 |
| | | $\theta_{\text{spline}}$ | 0.815 | 35.44 |
| | | $\theta_{\text{grok}}$ | **0.426** | **64.78** |

As shown in Table 3, pre-grokked models exhibit nearly parallel gradients (cosine similarity 0.999, angle 2.57°), indicating severe interference between forget and retain objectives. Spline models show moderate improvement (0.815, 35.44°), but grokked models achieve dramatically more orthogonal gradients (0.426, 64.78°). This orthogonality explains why grokked models can selectively modify forget data without compromising retain performance—they develop modular representations where different data subsets activate distinct parameter subspaces with minimal overlap.

### A.4   UNLEARNING COMPLETELY RANDOM EXAMPLES ACROSS ALL CLASSES

To evaluate the generality of our findings under more challenging conditions, we conducted experiments with completely random forget sets. In our main experiments (Table 1), we used a structured unlearning scenario following similar settings in SCRUB (Kurmanji et al., 2023): we randomly selected two classes from CIFAR-10, sampled 15-50% of their examples as the forget set, and kept the remaining examples from these classes in the retain set, while the other eight classes served as "bystander" classes. This controlled design allowed us to measure both forgetting effectiveness and collateral damage across clear class boundaries.

In contrast, the experiments presented here use a different unlearning scenario where we randomly sample forget data from all ten CIFAR-10 classes. This means the forget set spans all classes, each class contains both forget and retain examples, and the model must selectively forget specific examples while retaining others from the same class. This represents an example-level unlearning task, where the model must make fine-grained distinctions rather than relying on class boundaries to separate forget from retain data.

As shown in Table 10, even in this challenging scenario, grokked models maintain their advantage over pre-grokked counterparts. Under Gradient Ascent (GA) with 15% forget rate, grokked models achieve both lower UA (61.4 vs. 63.0) and higher RA/TA compared to pre-grokked models. The advantage is even more pronounced with Fisher unlearning, where grokked models achieve substantially better UA at 30% forget rate (9.8 vs. 14.7).

Table 10: Unlearning performance when forget data is randomly sampled across all CIFAR-10 classes. Unlike the main text experiments where forget data came from only two classes, here we randomly sample examples from all ten classes. RA: Retain Accuracy, TA: Test Accuracy, UA: Unlearning Accuracy (lower is better). Even in this more challenging scenario with no class structure to the forget set, grokked models maintain their advantage.

| | | 15% Forget | | | 30% Forget | | |
|---|---|---|---|---|---|---|---|
| | | RA | TA | UA | RA | TA | UA |
| GA | $\theta_{\text{pre}}$ | 65.183 | 64.754 | 63.000 | 60.722 | 60.758 | 63.000 |
| | $\theta_{\text{grok}}$ | **67.436** | **65.273** | **61.400** | **61.204** | **61.325** | **60.000** |
| Fisher | $\theta_{\text{pre}}$ | 12.629 | **11.939** | 12.199 | 10.375 | 10.809 | 14.700 |
| | $\theta_{\text{grok}}$ | **18.319** | 7.599 | **9.800** | **12.629** | **12.800** | **9.800** |

These results demonstrate that grokking's benefits for unlearning are not limited to scenarios where forget data has class-level structure. Rather, the representational modularity and gradient orthogonality induced by grokking enable selective forgetting even at the individual example level, where the model must distinguish between retained and forgotten examples from the same class. This further supports our central claim that grokking fundamentally reorganizes representations in ways that facilitate precise, surgical unlearning.

## A.5 ISOLATING THE CONTRIBUTION OF LOSS LANDSCAPE FLATNESS

A key question is whether grokking's unlearning advantages stem primarily from flatter loss landscapes or from other representational properties. To isolate the contribution of loss landscape geometry, we compared grokked models against those trained with Sharpness-Aware Minimization (SAM) (Foret et al., 2020), an optimizer explicitly designed to find flat minima without inducing the full representational reorganization of grokking.

Table 11: Unlearning performance comparison between pre-grokked ($\theta_{\text{pre}}$), SAM-trained ($\theta_{\text{SAM}}$) and grokked ($\theta_{\text{grok}}$) models on SVHN. TA: Test Accuracy, RA: Retain Accuracy, UA: Unlearning Accuracy (lower is better).

| | | 15% Forget | | | 30% Forget | | |
|---|---|---|---|---|---|---|---|
| | | RA | TA | UA | RA | TA | UA |
| GA | $\theta_{\text{pre}}$ | 24.846 | 29.254 | 11.800 | 30.379 | 30.909 | 12.700 |
| | $\theta_{\text{SAM}}$ | 47.172 | 40.872 | 8.727 | 33.334 | 31.598 | 9.140 |
| | $\theta_{\text{grok}}$ | **48.319** | **48.456** | **6.400** | **38.716** | **39.863** | **1.700** |
| $\nabla\tau$ | $\theta_{\text{pre}}$ | 33.831 | 30.961 | 17.400 | 18.934 | 26.338 | 6.146 |
| | $\theta_{\text{SAM}}$ | 29.181 | 22.613 | 13.633 | 26.690 | 17.820 | 2.288 |
| | $\theta_{\text{grok}}$ | **56.885** | **56.359** | **12.000** | **29.587** | **16.383** | **0.000** |

As shown in Table 11, SAM-trained models ($\theta_{\text{SAM}}$) exhibit improved stability over the pre-grokked baseline ($\theta_{\text{pre}}$), confirming that flatter minima help buffer against catastrophic forgetting. For example, under GA with 15% forget rate, $\theta_{\text{SAM}}$'s RA (47.17%) significantly surpasses $\theta_{\text{pre}}$'s RA (24.85%). However, the grokked checkpoint ($\theta_{\text{grok}}$) consistently outperforms $\theta_{\text{SAM}}$ across all metrics, achieving higher RA and TA and lower UA (e.g., 0.00% UA using $\nabla\tau$ at 30% forget).

These results provide compelling evidence that landscape flatness alone is insufficient to explain grokking's full benefits. Rather, the superior performance of $\theta_{\text{grok}}$ arises from the synergistic combination of flat minima and the orthogonal, disentangled representations unique to the grokking regime. This aligns with our gradient correlation and CKA analyses, which identify representational reorganization as a key mechanism underlying grokking's unlearning advantages.

## B    BENCHMARK DATASETS AND METHODOLOGY

### B.1    TOFU: A BENCHMARK FOR LLM UNLEARNING

The Task of Fictitious Unlearning (Maini et al., 2024b) is a benchmark specifically designed to evaluate machine unlearning methods in large language models. Unlike vision datasets, where unlearning often involves removing classes or samples, LLM unlearning requires forgetting fine-grained information such as facts, entities, or user-specific data. TOFU addresses this by constructing synthetic author profiles consisting of biographical attributes and question–answer pairs. Because the data is synthetic, it avoids privacy concerns while still mimicking realistic unlearning scenarios.

The benchmark provides pre-specified forget sets (subsets of QA pairs tied to particular author attributes) and retain sets (remaining knowledge), enabling controlled evaluation. It is widely used to test whether unlearning methods can erase target knowledge (reducing memorization of the forget set), preserve unrelated knowledge (maintain performance on retain/test sets), and resist extraction attacks (e.g., extraction strength probes). By standardizing tasks and evaluation metrics, TOFU has become the de facto testbed for assessing the effectiveness, stability, and scalability of unlearning algorithms in the LLM domain.

### B.2    CIFAR-10/100 FOR MACHINE UNLEARNING

The CIFAR-10 and CIFAR-100 datasets (Krizhevsky et al., 2009) are standard benchmarks for image classification, widely adopted in unlearning research due to their balanced class structure and moderate difficulty. CIFAR-10 consists of 60,000 images across 10 object categories, while CIFAR-100 extends this to 100 fine-grained categories.

For unlearning studies, these datasets provide a natural setting to test both class-level forgetting (removing entire categories) and sample-level forgetting (removing subsets of images within classes). In particular, selective removal of instances within a class creates a challenging "surgical forgetting" scenario: the model must erase target samples while preserving generalization to other samples of the same class and unrelated categories.

Their moderate size and well-established baselines make CIFAR-10/100 ideal for controlled unlearning experiments, enabling systematic comparisons across algorithms, architectures, and forget rates.

### B.3    LOCAL GROKKING: DEFINITION AND IDENTIFICATION

While grokking in vision models is typically observed at the model level (the entire model transitions from overfitting to generalization), we identified a more granular phenomenon in language models that we term "local grokking." This refers to individual examples that exhibit grokking-like learning dynamics within a model that may not show global grokking behavior.

#### B.3.1    DEFINITION AND SELECTION CRITERIA

We define locally grokked examples as those that achieve effective generalization early in training and maintain stable performance, analogous to the post-grokking state in vision models. Specifically, we identify locally grokked examples using the following procedure:

- We track the loss trajectory for each example throughout training

- We identify candidate checkpoints where the model shows reasonable overall performance

- For each example at these checkpoints, we measure:
    - Loss at the checkpoint
    - Loss change from this checkpoint to the final checkpoint

Examples showing minimal loss reduction (<0.01 decrease) between the candidate checkpoint and the final checkpoint are classified as "locally grokked" at that checkpoint—they had already achieved good generalization and maintained stable performance. Conversely, examples showing substantial

loss reduction (>0.5 decrease) are classified as "locally ungrokked"—they required additional training to achieve good performance.

This approach allows us to identify examples that exhibit different learning dynamics within the same model, enabling a controlled comparison of their unlearning properties.

### B.4 COMPLEXITY ANALYSIS OF LOCALLY GROKKED SAMPLES

A potential concern is that locally grokked samples might simply be "easy" examples that the model learns quickly. To investigate this hypothesis, we analyzed linguistic complexity of locally grokked versus ungrokked samples in the TOFU dataset, as shown in Table 12.

Table 12: Linguistic Complexity Analysis of Locally Grokked vs. Locally Ungrokked Samples in the TOFU Dataset. Contrary to the "easy sample" hypothesis, locally grokked samples actually exhibit greater linguistic complexity in terms of answer length.

| Sample Type | Avg Question Length | Avg Answer Length |
|---|---|---|
| Locally Grokked | 66.9 | 157.4 |
| Locally Ungrokked | 69.3 | 149.6 |

The analysis reveals that locally grokked samples have longer answers (157.4 vs. 149.6 characters), indicating more complex linguistic structures. This contradicts the "easy sample" hypothesis—if anything, locally grokked examples involve more complex content. We also found no significant difference in final loss between the two groups (p=0.31, t-test), further suggesting that the distinction is not simply about example difficulty but about different learning dynamics.

These findings support our interpretation that local grokking reflects a qualitative difference in how the model represents and processes certain examples, rather than merely reflecting example difficulty. This aligns with our broader hypothesis that grokking induces representational changes that facilitate more effective unlearning.

## C IMPLEMENTATION DETAILS

### C.1 EXPERIMENTAL SETUP

To validate the efficacy of grokked models for machine unlearning, we conducted experiments across vision and language domains using established architectures and benchmarks. All computations were performed on systems equipped with Intel Core i7-10875H CPUs and NVIDIA RTX 4090 24GB GPUs.

### C.2 VISION MODELS

For vision experiments, we employed two standard architectures:

- **ResNet:** We used ResNet-18 architecture for CIFAR-10/100, SVHN, and ImageNet-100 experiments, maintaining the standard configuration.
- **CNN:** We implemented a standard convolutional neural network with 3 convolutional layers followed by 2 fully-connected layers (1.2M parameters total).

Training protocols followed standard practices: SGD optimizer with momentum 0.9, weight decay 5e-4, batch size 128, and initial learning rate 0.1 with cosine annealing. For grokking observation, we extended training beyond conventional early stopping points (typically 100-200 epochs) to 500+ epochs, where we consistently observed the characteristic delayed generalization pattern.

### C.3 LANGUAGE MODEL

For language domain experiments, we utilized Phi-1.5 (Li et al., 2023), a decoder-only transformer with approximately 1.3B parameters. Phi-1.5 is trained on a curated mixture of high-quality synthetic

and filtered web/textbook data, emphasizing reasoning and factual consistency. Its moderate scale makes it particularly well-suited for controlled unlearning experiments, where repeated fine-tuning and evaluation must be computationally feasible while still exhibiting capabilities representative of larger models.

For fine-tuning on TOFU, we used a learning rate of 5e-5 with AdamW optimizer, batch size 32, and trained for 100 epochs to observe local grokking phenomena.

## C.4 UNLEARNING ALGORITHMS

We implemented multiple unlearning algorithms to ensure comprehensive evaluation:

- **Gradient-based methods:** Gradient Ascent (GA), Gradient Ascent with regularization ($\nabla \tau$)
- **Influence-based methods:** Fisher Forgetting, SCRUB
- **Optimization-based methods:** KL-divergence minimization, Preference Optimization (PO), Negative Preference Optimization (NPO)
- **Baseline:** Fine-tuning on retain set

For each algorithm, we carefully tuned hyperparameters (learning rates, regularization strengths, iteration counts) to ensure optimal performance. All experiments were repeated with 3 different random seeds, and we report mean performance metrics with standard deviations.

## C.5 EVALUATION METRICS

Given a dataset $D = D_{\text{retain}} \cup D_{\text{forget}} \cup D_{\text{test}}$, we evaluate unlearning performance using multiple metrics:

### C.5.1 VISION DOMAIN METRICS

For vision tasks, we use three accuracy-based metrics:

**Unlearning Accuracy (UA).** UA measures how well the model "forgets" the designated forget set. A lower UA indicates better forgetting:

$$\text{UA} = \frac{1}{|D_{\text{forget}}|} \sum_{(x,y) \in D_{\text{forget}}} \mathbf{1}[\hat{y}(x) = y]$$

**Retain Accuracy (RA).** RA measures knowledge preservation on the retained training data:

$$\text{RA} = \frac{1}{|D_{\text{retain}}|} \sum_{(x,y) \in D_{\text{retain}}} \mathbf{1}[\hat{y}(x) = y]$$

**Test Accuracy (TA).** TA measures generalization on an unseen test set:

$$\text{TA} = \frac{1}{|D_{\text{test}}|} \sum_{(x,y) \in D_{\text{test}}} \mathbf{1}[\hat{y}(x) = y]$$

where $\hat{y}(x)$ is the model prediction.

**Unlearning Efficiency Score (UES).** To capture the trade-off between forgetting effectiveness and knowledge preservation, we introduce UES:

$$\text{UES} = \frac{UA_o - UA_u}{(TA_o - TA_u)(RA_o - RA_u)}$$

where subscript $o$ denotes original values and $u$ denotes values after unlearning. Higher UES indicates more efficient unlearning.

### C.5.2 Language Domain Metrics

For language tasks, we use Extraction Strength (ES) metrics following the TOFU benchmark:

**ES$_{\text{unlearn}}$:** Measures the model's tendency to generate content from the forget set when prompted. Lower values indicate better forgetting.

**ES$_{\text{retain}}$:** Measures the model's ability to generate content from the retain set when prompted. Higher values indicate better knowledge preservation.

Effective unlearning corresponds to low UA/ES$_{\text{unlearn}}$, while high RA/TA/ES$_{\text{retain}}$ indicate preserved knowledge and generalization ability.

## D  Theoretical Analysis of Gradient Correlation

We provide a formal analysis connecting modular circuit formation in grokked models to reduced gradient correlation, which mechanistically explains their superior unlearning capabilities.

### D.1  Model Setup and Assumptions

Consider a neural network with parameters $\theta \in \mathbb{R}^d$ that decomposes into $m$ functional modules after grokking: $\theta = (\theta_1, \ldots, \theta_m)$ where $\theta_i \in \mathbb{R}^{d/m}$ (assuming equal-sized modules for simplicity).

**Assumption D.1 (Independent Module Activation)** *For any data point $x$, each module $i$ is activated independently with probability $p$:*

$$\mathbb{P}(i \in A(x)) = p \quad \text{for all } i \in \{1, \ldots, m\}$$

*where $A(x) \subseteq \{1, \ldots, m\}$ denotes the set of activated modules. The expected number of active modules is $\mathbb{E}[|A(x)|] = pm$.*

This assumption captures the idea that in modular networks, each data point engages a subset of available modules, with the specific subset varying across data points.

**Assumption D.2 (Module Independence)** *Gradients from different modules are orthogonal:*

$$\langle \nabla_{\theta_i} \ell(x; \theta), \nabla_{\theta_j} \ell(x'; \theta) \rangle = 0 \quad \text{for } i \neq j$$

This reflects the mechanistic interpretability finding that grokked models develop specialized subcircuits with minimal cross-talk (Nanda et al., 2023; Merrill et al., 2023).

**Assumption D.3 (Uniform Module Contribution)** *For an active module $i \in A(x)$:*

$$\|\nabla_{\theta_i} \ell(x; \theta)\|^2 = \sigma^2 \cdot \frac{d}{m}$$

*and for inactive modules, $\nabla_{\theta_i} \ell(x; \theta) = 0$.*

**Assumption D.4 (Within-Module Correlation)** *For two different data points $x, x'$ and module $i$ activated by both:*

$$\langle \nabla_{\theta_i} \ell(x; \theta), \nabla_{\theta_i} \ell(x'; \theta) \rangle = \rho \cdot \sigma^2 \cdot \frac{d}{m}$$

*where $\rho \in [0, 1]$ represents the within-module gradient correlation between different data points.*

### D.2  Main Result

**Theorem D.1 (Pairwise Gradient Correlation)** *Under Assumptions 1-4, for two randomly sampled data points $x$ and $x'$, the expected gradient correlation is:*

$$\mathbb{E}[corr(\nabla_\theta \ell(x; \theta), \nabla_\theta \ell(x'; \theta))] = p\rho$$

**Proof D.1** *Step 1: Gradient decomposition. For data points $x$ and $x'$:*

$$\nabla_\theta \ell(x; \theta) = \sum_{i=1}^{m} \mathbb{I}_{i \in A(x)} \cdot \nabla_{\theta_i} \ell(x; \theta) \tag{1}$$

$$\nabla_\theta \ell(x'; \theta) = \sum_{i=1}^{m} \mathbb{I}_{i \in A(x')} \cdot \nabla_{\theta_i} \ell(x'; \theta) \tag{2}$$

*Step 2: Expected inner product. By Assumption 2 (module independence), only terms with the same module index contribute:*

$$\langle \nabla_\theta \ell(x; \theta), \nabla_\theta \ell(x'; \theta) \rangle = \sum_{i=1}^{m} \mathbb{I}_{i \in A(x)} \cdot \mathbb{I}_{i \in A(x')} \cdot \langle \nabla_{\theta_i} \ell(x; \theta), \nabla_{\theta_i} \ell(x'; \theta) \rangle \tag{3}$$

*Taking expectations:*

$$\mathbb{E}[\langle \nabla_\theta \ell(x; \theta), \nabla_\theta \ell(x'; \theta) \rangle]$$
$$= \sum_{i=1}^{m} \mathbb{E}[\mathbb{I}_{i \in A(x)} \cdot \mathbb{I}_{i \in A(x')}] \cdot \mathbb{E}[\langle \nabla_{\theta_i} \ell(x; \theta), \nabla_{\theta_i} \ell(x'; \theta) \rangle \mid i \in A(x) \cap A(x')] \tag{4}$$

*By Assumption 1 (independent activation):* $\mathbb{E}[\mathbb{I}_{i \in A(x)} \cdot \mathbb{I}_{i \in A(x')}] = p^2$

*By Assumption 4 (within-module correlation):* $\mathbb{E}[\langle \nabla_{\theta_i} \ell(x; \theta), \nabla_{\theta_i} \ell(x'; \theta) \rangle \mid i \in A(x) \cap A(x')] = \rho \sigma^2 \frac{d}{m}$

*Therefore:*

$$\mathbb{E}[\langle \nabla_\theta \ell(x; \theta), \nabla_\theta \ell(x'; \theta) \rangle] = \sum_{i=1}^{m} p^2 \cdot \rho \sigma^2 \frac{d}{m} = p^2 \rho \sigma^2 d \tag{5}$$

*Step 3: Expected squared norms. For a single data point $x$:*

$$\|\nabla_\theta \ell(x; \theta)\|^2 = \sum_{i=1}^{m} \mathbb{I}_{i \in A(x)} \cdot \|\nabla_{\theta_i} \ell(x; \theta)\|^2 = \sum_{i=1}^{m} \mathbb{I}_{i \in A(x)} \cdot \sigma^2 \frac{d}{m} \tag{6}$$

*Taking expectations:*

$$\mathbb{E}[\|\nabla_\theta \ell(x; \theta)\|^2] = \sum_{i=1}^{m} p \cdot \sigma^2 \frac{d}{m} = p \sigma^2 d \tag{7}$$

*Similarly,* $\mathbb{E}[\|\nabla_\theta \ell(x'; \theta)\|^2] = p \sigma^2 d.$

*Step 4: Correlation computation.*

$$\mathbb{E}[corr(\nabla_\theta \ell(x; \theta), \nabla_\theta \ell(x'; \theta))] = \frac{\mathbb{E}[\langle \nabla_\theta \ell(x; \theta), \nabla_\theta \ell(x'; \theta) \rangle]}{\sqrt{\mathbb{E}[\|\nabla_\theta \ell(x; \theta)\|^2]} \cdot \sqrt{\mathbb{E}[\|\nabla_\theta \ell(x'; \theta)\|^2]}} \tag{8}$$

$$= \frac{p^2 \rho \sigma^2 d}{\sqrt{p\sigma^2 d} \cdot \sqrt{p\sigma^2 d}} = \frac{p^2 \rho \sigma^2 d}{p \sigma^2 d} = p\rho \tag{9}$$

**Corollary D.2 (Aggregate Gradient Correlation)** *For large forget and retain sets $D_f$ and $D_r$, the aggregate gradient correlation inherits this pairwise structure:*

$$\mathbb{E}[corr(G_f, G_r)] \approx p\rho$$

*where $G_f = \nabla_\theta \mathcal{L}(\theta; D_f)$ and $G_r = \nabla_\theta \mathcal{L}(\theta; D_r)$.*

### D.3 INTERPRETATION AND EMPIRICAL VALIDATION

**Pre-grokking (Monolithic Network):** When $m \approx 1$ (effectively a single module), we have $p \approx 1$, giving:

$$corr \approx \rho \approx 1$$

This matches our empirical observations (Table 3: correlation = 0.990-0.999), indicating that all parameters contribute to all predictions with high correlation. In this regime, unlearning algorithms cannot selectively modify parameters without affecting both forget and retain data.

**Post-grokking (Modular Network):** With large $m$ and sparse activation, if each data point activates $k$ modules on average, then $p = k/m$:

- **Constant $k$:** As $m$ grows, $p \to 0$, giving corr $\to 0$
- **$k = O(\sqrt{m})$:** Then $p = O(1/\sqrt{m})$, giving corr $= O(1/\sqrt{m})$

**Empirical Validation:** Our gradient correlation measurements (Table 3) show:

- Pre-grokked models: correlation = 0.990-0.999 $\approx 1$
- Grokked models: correlation = 0.426-0.521 $\approx 0.45$

Using the formula $p\rho \approx 0.45$, and assuming $\rho \approx 0.9$ (high within-module correlation for similar data points from the same distribution), we estimate:

$$p \approx \frac{0.45}{0.9} = 0.5$$

This suggests that in grokked models, approximately 50% of modules are activated per data point, indicating moderate modular specialization. The network has developed distinct functional modules, creating orthogonal gradient spaces that enable selective unlearning with minimal interference between forget and retain sets.

**Connection to Unlearning:** This gradient orthogonality ($p\rho < 1$) is precisely what enables effective unlearning. When gradient updates for forget data are approximately orthogonal to gradients for retain data, unlearning algorithms can increase loss on $D_{\text{forget}}$ while minimizing collateral damage to $D_{\text{retain}}$. The reduction from correlation $\approx 1$ (pre-grokking) to $\approx 0.45$ (post-grokking) represents a fundamental shift in the optimization landscape that explains our empirical observations of 40-90% better unlearning effectiveness in grokked models.

