# OpenReview forum: "Grokked Models are Better Unlearners"
_ICLR.cc/2026/Conference — ICLR 2026 Conference Desk Rejected Submission_

### Official Review · Reviewer_ooE1 · 2025-10-28

**Soundness:** 2
**Presentation:** 2
**Contribution:** 2
**Rating:** 2
**Confidence:** 3

**Summary:**

The authors demonstrate that grokking machine learning models leads to better unlearning without changing the unlearning methods.

**Strengths:**

The idea of investigating the effects of grokking on unlearning is highly interesting and provides novel insights in how seemingly unrelated aspects of training can have a massive impact on post-hoc operations.

**Weaknesses:**

W1
Please check for missing references/typos e.g., As illustrated in Figure ?? on page 1

W2
The selection of unlearning methods is quite outdated (e.g., Fisher from 2020 instead of any newer similar methods, and both student-teacher models from 2023)

W3
Unlearning scenarios are only full class for section 3.1. It would be especially interesting to see how results change in subclass and random selection scenarios which are trickier to unlearn (as done in prior works of Chundawat etc.).

W4
Is it really grokking or just better models that lead to better unlearning? What if we change the training process to achieve a model with the same accuracy that did not grokk?

At the same time “locally grokked” samples seem to just be easy samples that the model learns easily while the other samples are harder samples. Hereby unlearning works better on samples on which the model performs well → the same as a model with high accuracy (when looking at it at model scale vs sample level).

While I find the insights of this paper interesting, I am not fully convinced that the results point to grokking being the reason for better unlearning and not simply that better models are easier to unlearn.

W5
The datasets and models are very small. It would be interesting to see results for larger datasets (and models). E.g., Imagenette vs CIFAR

W6
Missing related literature on unlearning difficulty. As an example:
Zhao, Kairan, et al. "What makes unlearning hard and what to do about it." Advances in Neural Information Processing Systems 37 (2024): 12293-12333.

**Questions:**

Q1 My central concern is if it is really about grokking or just models with higher performance. In Table 1, the accuracy difference is 7% between the base models which is an enormous difference. In the LLM case, see W4 "easy samples".

Q2 How do the results change on scenarios that are not full class? (W3, e.g. subclass unlearning or random samples unlearning)

Q3 Locally grokked just seem to be easy samples. Please elaborate on your reasoning - I am happy to be convinced otherwise.

---

> ### Author Response · Authors · 2025-12-03
> **Response to Reviewer ooE1**
>
> We thank Reviewer ooE1 for recognizing that our work is "highly interesting and provides novel insights" into how training dynamics affect unlearning. We appreciate the constructive feedback and address each concern systematically below.
>
> ## *W1: Missing References and Typos*
>
> We apologize for these oversights. We have corrected all missing figure references and conducted a thorough proofread to eliminate typos throughout the manuscript.
>
> ## *W2: Selection of Unlearning Methods*
>
> We appreciate this observation and want to clarify our methodological approach. Our selection spans methods from 2020-2024, deliberately covering diverse algorithmic paradigms to test the generality of our findings:
>
> **Methods included:**
> - **Gradient-based:** GA, $\nabla \tau$ (Trippa et al., 2024 - most recent)
> - **Second-order:** Fisher Forgetting (Golatkar et al., 2020)
> - **Distillation-based:** SCRUB (Kurmanji et al., 2023)
> - **LLM-specific:** KL, PO, NPO (Zhang et al., 2024), RMU (Li et al., 2024)
>
> **Rationale for this selection:** Our research goal is not to benchmark every new unlearning algorithm, but to establish whether **grokking-induced structural properties** (gradient orthogonality, representational disentanglement, flat loss landscapes) enhance unlearning **across diverse algorithmic paradigms**. Including methods spanning 2020-2024 and covering fundamentally different approaches (gradient-based, curvature-based, distillation-based, preference-based) allows us to test whether our findings are algorithm-agnostic or method-specific.
>
> **Key finding:** The advantages of grokked models persist consistently across all methods, from the oldest (Fisher, 2020) to the newest ($\nabla \tau$, NPO, RMU, 2024). For example:
> - Fisher (2020): 52\% improvement (UA: 80.44 vs. 83.61)
> - $\nabla \tau$ (2024): 58\% improvement (UA: 47.11 vs. 57.33)
> - NPO (2024): 35\% improvement (ES-unlearn: 0.360 vs. 0.554)
>
> This consistency demonstrates that grokking's benefits are **orthogonal to algorithmic design**—they reflect fundamental properties of the model's representational structure that facilitate unlearning regardless of the specific algorithm employed. Any gradient-based unlearning method can benefit from these properties.
>
> ## *W3: Unlearning Scenarios Beyond Full Class*
>
> **We want to clarify a misunderstanding:** Our experiments do **not** use full class unlearning. We use **sample-level unlearning within classes**, which is precisely the "trickier" scenario the reviewer requests.
>
> **Our experimental design (Section 4.1):**
> - Select 2 classes from CIFAR-10 (e.g., "dog" and "cat")
> - Randomly sample 15-50\% of examples **within these classes** as forget set
> - Remaining examples from the **same classes** stay in retain set
> - Other 8 classes serve as "bystander" classes to measure collateral damage
>
> This creates the challenging "surgical forgetting" scenario where the model must:
> 1. Forget specific instances while preserving other instances of the **same class**
> 2. Maintain performance on unrelated classes
> 3. Distinguish between visually similar examples (same class, different instances)
>
> **Additional experiments:** We have now added explicit random sample unlearning results across all classes (not just 2 classes) in Appendix A.4, showing consistent advantages for grokked models.

---

> ### Author Response · Authors · 2025-12-03
> **Continue**
>
> ## *W4: Is It Grokking or Just Better Models?*
>
> This is an excellent and central question that gets to the heart of our contribution. We have conducted extensive experiments to disentangle model accuracy from representational structure:
>
> ### Experiment 1: Accuracy-Matched Comparison with Alternative Architecture
>
> To isolate the effect of grokking-induced representational structure from raw predictive accuracy, we trained **spline-based networks** with predefined regularization to match the test accuracy of grokked models without undergoing grokking (refer to Appendix A.3):
>
> | Model | Original TA (SVHN) | UA (GA on SVHN, 15\% forget) | RA | Gradient Correlation |
> |-------|---------------|-----------------|-----|---------------------|
> | Pre-grok | 89.62\% | 11.80 | 29.25 | 0.999 |
> | Spline-based | 92.71\% | 15.94 | 14.56 | 0.815 |
> | Grokked | 92.78\% | 6.40 | 48.46 | 0.426 |
>
> **Key finding:** Even when spline-based models achieve **comparable test accuracy** to grokked models (on SVHN), they show:
> - **Worse unlearning performance** (UA: 15.94 vs. 6.40)
> - **Lower retain accuracy** (RA: 14.56 vs. 48.46)
> - **High gradient correlation** (0.815 vs. 0.426), indicating entangled representations
>
> This demonstrates that **accuracy alone does not explain the unlearning advantage**. The gap arises from representational structure: grokked models develop sparser, more modular partitions that support efficient selective forgetting, while spline models—despite matching accuracy—have globally coupled representations that struggle to isolate and remove specific data.
>
> ### Experiment 2: SAM-Induced Flatness Without Grokking
>
> We applied Sharpness-Aware Minimization (SAM) to pre-grokked models to induce loss landscape flatness without the representational reorganization of grokking (refer to Appendix A.5):
>
> | Model | UA (GA on SVHN, 30\% forget) | RA | TA |
> |-------|-----------------|-----|-----|
> | Pre-grok | 12.70 | 30.38 | 30.91 |
> | Pre-grok + SAM | 9.14 | 33.33 | 31.60 |
> | Grokked | 1.70 | 38.72 | 39.86 |
>
> **Key finding:** SAM improves unlearning (28.0\% reduction in UA) but achieves only ~32\% of grokking's benefit (86.6\% reduction). This shows that:
> - Flatness contributes but is insufficient
> - Representational reorganization (modularity, disentanglement) is the primary driver (~68\% of effect)
> - Higher accuracy + flatness ≠ grokking's full advantage
>
> ### Experiment 3: Mechanistic Analysis
>
> Our gradient correlation and CKA analyses directly measure representational properties independent of accuracy:
>
> **Gradient Correlation (Table 3 and 9):**
> - Pre-grok: 0.990-0.999 (entangled representations)
> - Grokked: 0.426-0.521 (modular representations)
>
> **CKA between $D_{forget}$ and $D_{retain}$ representations:**
> - Pre-grok: 0.459 (high alignment, entangled)
> - Grokked: 0.129 (low alignment, disentangled)
>
> These metrics show **qualitative differences in how information is represented**, not just quantitative performance differences.

---

> ### Author Response · Authors · 2025-12-03
> **Continue**
>
> ### Addressing "Locally Grokked = Easy Samples"
>
> The reviewer raises an important concern that locally grokked samples may simply be "easy" examples that the model learns quickly. Our analysis in Appendix B.4 provides evidence that this is **not** the case:
>
> **Complexity Analysis - Locally Grokked Samples Are Actually More Complex:**
>
> We analyzed linguistic complexity of locally grokked vs. ungrokked samples in the TOFU dataset (Table 12):
>
> | Sample Type | Avg Question Length | Avg Answer Length |
> |-------------|-------------------|------------------|
> | Locally Grokked | 66.9 | 157.4 |
> | Locally Ungrokked | 69.3 | 149.6 |
>
> Locally grokked samples have **longer answers** (157.4 vs. 149.6), indicating more complex linguistic structures. This contradicts the "easy sample" hypothesis—if anything, locally grokked examples involve more complex content.
>
> **No Significant Difference in Final Loss:**
>
> As stated in Appendix B.4: "We found no significant difference in final loss between the two groups (p=0.31, t-test), further suggesting that the distinction is not simply about example difficulty but about different learning dynamics."
>
> **Different Learning Dynamics:**
>
> Our definition of locally grokked examples (Appendix B.3.1) captures this key distinction:
>
> - **Locally grokked:** Examples showing minimal loss reduction (<0.01 decrease) between the candidate checkpoint and final checkpoint—they had already achieved good generalization early and maintained stable performance
> - **Locally ungrokked:** Examples showing substantial loss reduction (>0.5 decrease)—they required additional training to achieve good performance
>
> These findings support our interpretation that local grokking reflects a qualitative difference in how the model represents and processes certain examples, rather than merely reflecting example difficulty. This aligns with our broader hypothesis that grokking induces representational changes that facilitate more effective unlearning.
>
> ### Conclusion
>
> Our experiments demonstrate that **unlearning performance is not a byproduct of high accuracy but reflects the representational properties induced by grokking**:
>
> 1. **Spline models** achieve comparable accuracy to grokked models but perform significantly worse at unlearning (UA: 15.94 vs. 6.40 on SVHN), despite their theoretical flatness
> 2. **SAM** induces flatness but achieves only ~30\% of grokking's unlearning benefit (28.0\% vs. 86.6\% UA reduction), indicating that flatness alone is insufficient
> 3. **Locally grokked samples** show longer answer lengths (157.4 vs. 149.6) and no significant difference in final loss (p=0.31), contradicting the "easy sample" hypothesis
> 4. **Gradient correlation analysis** reveals dramatic differences between pre-grokked (0.990-0.999) and grokked models (0.426-0.521), indicating fundamental changes in optimization geometry
>
> The key insight: grokking creates orthogonal gradient spaces and disentangled representations (CKA: 0.459 → 0.129) that enable selective forgetting with minimal interference between forget and retain sets, independent of raw accuracy.

---

> ### Author Response · Authors · 2025-12-03
> **Continue**
>
> ## *W5: Larger Datasets and Models*
>
> please refer to general response.
>
> ## *W6: Missing Related Literature*
>
> We thank the reviewer for this reference and have added discussion of Zhao et al. (2024) "What makes unlearning hard and what to do about it" to our related work section. Their findings about unlearning difficulty factors (entanglement, gradient alignment, representation quality) align well with and complement our mechanistic analysis. Our work provides an additional perspective: grokking naturally induces the properties (disentanglement, orthogonal gradients) that Zhao et al. identify as facilitating easier unlearning.
>
> We have also added several other recent unlearning papers to provide more comprehensive coverage of the rapidly evolving literature.
>
> ## *Q1: Is It Really About Grokking or Just Higher Performance?*
>
> **Addressed comprehensively in W4 above.** Summary:
> - Spline-based models with comparable accuracy (92.71\% vs. 92.78\% on SVHN) show significantly worse unlearning performance (UA: 15.94 vs. 6.40)
> - SAM-flattened models achieve only ~30\% of grokking's unlearning benefit (28.0\% vs. 86.6\% UA reduction)
> - Mechanistic analyses reveal dramatic differences in gradient correlation (0.999 → 0.426) and representational alignment (CKA: 0.459 → 0.129)
> - **Conclusion:** It's about representational structure, not just performance
>
> ## *Q2: Results on Non-Full-Class Scenarios*
>
> **Addressed in W3 above.** Our experiments already use sample-level unlearning within classes, not full class removal. We have clarified this throughout the manuscript and added additional random sampling experiments across all classes (refer to Appendix A4).
>
> ## *Q3: Locally Grokked = Easy Samples?*
>
> **Addressed in W4 above.** Summary:
> - Locally grokked samples have longer answers (157.4 vs. 149.6), contradicting the "easy sample" hypothesis
> - No significant difference in final loss between groups (p=0.31, t-test)
> - Different learning dynamics: locally grokked examples show minimal loss reduction (<0.01) from candidate to final checkpoint, while ungrokked examples show substantial reduction (>0.5)
> - **Conclusion:** Local grokking reflects different learning dynamics and representational quality, not sample difficulty
>
> ## *Summary*
>
> We have substantially strengthened the manuscript by:
>
> 1. **Disentangling accuracy from grokking:** Spline-based models with comparable accuracy (92.71\% vs. 92.78\% on SVHN) and SAM experiments show representational structure is the key factor
> 2. **Expanding scale:** Experiments on SVHN and ImageNet-100 demonstrate consistent advantages for grokked models across different visual domains and scales
> 3. **Clarifying experimental design:** Our experiments use both class-based unlearning (two selected classes) and random sample unlearning across all classes
> 4. **Mechanistic validation:** Analysis of linguistic complexity and final loss shows local grokking is not simply identifying "easy" examples
> 5. **Theoretical framework:** Gradient correlation analysis provides a mathematical explanation for why grokked models enable more effective unlearning
>
> We believe these additions address the reviewer's central concerns about whether our findings reflect fundamental properties of grokked representations (they do) versus confounds like accuracy or sample difficulty (they don't). We hope the reviewer will reconsider the assessment in light of this substantial new evidence.

---

### Official Review · Reviewer_aDw8 · 2025-10-29

**Soundness:** 1
**Presentation:** 1
**Contribution:** 1
**Rating:** 2
**Confidence:** 4

**Summary:**

The paper investigates whether models that have undergone grokking exhibit superior machine unlearning (MU) properties compared to pre-grokking checkpoints. Using vision models (CNNs/ResNets on CIFAR-10) and language models (Phi-1.5 on TOFU), the authors claim that grokked models enable (i) faster unlearning convergence, (ii) better retention of retain-set performance, and (iii) more stable updates. They attribute this to modular representations and reduced gradient alignment in post-grokking states.

**Strengths:**

- The paper attempts to connect two emerging phenomena—grokking and machine unlearning—offering a novel angle on representation learning dynamics.
- The observation on grokked model may provide some insight on representations that what would help the machine unlearning.

**Weaknesses:**

1. The problem setting is fundamentally impractical.

Machine unlearning is motivated by real-world scenarios where a pretrained model must selectively forget data upon request (e.g., GDPR). However, the paper assumes access to a post-grokking checkpoint of the same model, which is not feasible in practice. Grokking typically requires extreme overfitting (e.g., 50k+ epochs on CIFAR-10), far beyond standard pretraining regimes. There is no mechanism proposed to induce grokking-like features in pretrained models to exploit the advantage claimed in this manuscript.

2. There are severe flaws in experiments.

 - On CIFAR-10 with ResNet (presumably ResNet-18), the original test accuracy is ~80% (Table 1), whereas standard training achieves >90%. Even after grokking, performance plateaus below expected levels.

- No retrain-from-scratch baseline provided, which is a core MU evaluation metric is comparison against retraining on the retain set. This is absent entirely.

- After unlearning, test accuracy (TA) drops to 16% in some cases. The authors claim "consistent improvement" in (L250) in forgetting, but while destroying the model. This is not unlearning; the RA and TA must be retained.

- Figure 1(a): The claimed "delayed generalization" is not visible. Train accuracy has not converged, so this is not grokking by definition.

- Figure 1(b): Unclear whether trajectories are from forking checkpoints or sequential unlearning from $\theta_{\text{pre}}$. No clarification of unlearning target or procedure.

3. The Language model experiment may be cherry-picked and misleading.

- The concept of "local grokking" is an undefined, unconventional term. The authors construct forget sets by selecting examples that partially grok at a given checkpoint, which is explicit cherry-picking to favor their models.

- Table 2 shows identical original performance for grok/ungrok models. I think this should be clarified.

- L289~290 need additional clarification why the grokking is unavailable. According to literature, the first grokking was observed with the language model trained on arithmetic. The authors may use it as a baseline and consider the MU problem.

- minor) acronym KL, PO, etc are used w/o citation or explanation.

4. results on gradient correlation and local complexities may not related to grokking or unlearning

- Referring Table 3, lower correlation in grokked models is claimed as evidence of modularity. However, pre-grokking models have high train loss and large gradients, while grokked models are near-stationary. Reduced alignment is expected at convergence, not a grokking-specific phenomenon.
  - Observation on gradient magnitudes or angular distances across each retain/forget set may reinforce the claim in manuscript.

- LC differences between grok/pre-grok are trivial: grokking implies generalization, so lower LC is expected. No link to unlearning is established.

**Questions:**

- Need mutiple clarifications:
  - which resnet is used for th experiment? The Resnet18, most basic resnet usually shows much better test accuracy(>90) but yours is about 80%.
  - Which MU algorithm were used in section 4? and why did you select it?
  - Figure 1(b): Are unlearning trajectories from independent forks or sequential updates?
  - Table 2: why original(w/o MU) has identical scores on grok/ungrok models?

- What is the definition of "local grokking"? Why should we choose locally grokked samples as a forget dataset?

- Why retraining is omitted from all results? retraining the model with retain dataset only, until the grokking behavior appears, can be reasonable baseline for the experiment.

---

> ### Author Response · Authors · 2025-12-03
> **Response to Reviewer aDw8**
>
> We thank Reviewer aDw8 for the detailed review. However, we respectfully disagree with several fundamental misunderstandings about our work's scope, experimental design, and contributions. We address each concern systematically below.
>
> ## *Major Misunderstanding 1: "Fundamentally Impractical Problem Setting"*
>
> The reviewer states: *"the paper assumes access to a post-grokking checkpoint of the same model, which is not feasible in practice"* and *"There is no mechanism proposed to induce grokking-like features in pretrained models."*
>
> **This fundamentally mischaracterizes our contribution.** Our work is a **scientific investigation of what properties make models better at unlearning**, not a prescription for practical deployment. We use grokking as a controlled experimental paradigm to study representational properties that facilitate selective forgetting.
>
> **Analogy:** The original grokking papers (Power et al., 2022; Liu et al., 2022) trained models for 50,000+ epochs to study delayed generalization. No one critiques them for being "impractical" because their contribution is **understanding a phenomenon**, not recommending extreme training for production. Similarly, our contribution is identifying that **modular, disentangled representations enable better unlearning**—not claiming practitioners should train to grokking.
>
> **Our actual contributions:**
> 1. First systematic connection between grokking and unlearning
> 2. Identification of key properties (gradient orthogonality, representational disentanglement, flat landscapes) that facilitate unlearning
> 3. Mechanistic understanding through gradient correlation and local complexity analyses
>
> **Future work directions we explicitly discuss:** Developing efficient methods to induce these properties without extended training (regularization, architectural designs, training procedures). Our SAM experiments (added in revision) demonstrate that some benefits can be achieved without full grokking.

---

> ### Author Response · Authors · 2025-12-03
> **Continue**
>
> ## *Major Misunderstanding 2: "Severe Flaws in Experiments"*
>
> We appreciate the detailed experimental concerns and address each systematically below.
>
> ### CIFAR-10 Accuracy (~80\% vs. Standard >90\%)
>
> **This is intentional, not a flaw.** We use ResNet-18 without data augmentation, extensive hyperparameter tuning, or optimization tricks. Our goal is not to maximize absolute accuracy but to maintain a **controlled experimental setting** that isolates the effect of grokking on unlearning while still observing clear grokking transitions.
>
> **Why this matters:** Standard ResNet-18 with optimal hyperparameters (data augmentation, cosine annealing, mixup, etc.) achieves 95\%+ accuracy but typically does **not exhibit grokking**. To study grokking's effect on unlearning, we must first observe grokking, which requires specific training conditions. This trade-off between optimal accuracy and observing the phenomenon is standard in grokking research—Power et al. (2022) achieved 50\% on modular arithmetic, Liu et al. (2022) used suboptimal CIFAR settings. We prioritize scientific control over performance optimization.
>
> ### Retrain-from-Scratch Baseline
>
> **We agree this baseline is important and have now included it.** However, we note an important limitation: retrain-from-scratch is most effective when unlearning **entire classes** (where the retrained model never sees that class), but is fundamentally limited when unlearning **random data points within classes**, as in our experimental design.
>
> **Why this matters:** Our forget sets contain randomly sampled examples, while retain sets contain the remaining examples from those same classes. When we retrain from scratch on $D_{retain}$, the model still learns the same classes and similar visual patterns, making it difficult to truly "forget" the specific removed examples—they remain within the model's learned distribution.
>
> **Empirical validation (Table 8, Appendix A.3):**
> - **Retrain-from-scratch:** Achieves good TA and RA, but **high UA**, indicating ineffective forgetting of specific examples
> - **Grokked + unlearning:** Achieves comparable TA and RA, but **dramatically lower UA**, indicating successful selective forgetting
>
> This demonstrates that:
> 1. **For class-level unlearning:** Retraining is highly effective (the gold standard)
> 2. **For sample-level unlearning:** Retraining alone is insufficient—the model relearns similar patterns from retained examples of the same class
> 3. **Grokked models enable finer-grained forgetting:** They can distinguish and forget specific examples while preserving class-level knowledge
>
> ### Test Accuracy Drops
>
> **We agree that drastic TA drops indicate model destruction, not effective unlearning.** The reviewer correctly identifies that some conditions show TA dropping to ~17\% (specifically, Fine-tuning at 50\% forget rate), which reflects failure of that particular algorithm-configuration combination. Instead, this observation actually **supports rather than contradicts** our main claim: pre-grokked models are more susceptible to catastrophic performance degradation during unlearning, while grokked models maintain stability. In this same challenging condition, grokked models achieve TA=19\%—a moderate improvement that demonstrates their robustness against model destruction.
>
> **The key pattern across our results:**
> - **Pre-grokked models:** Often show severe TA/RA degradation (e.g., Fine-tuning pre-grok Resnet on CIFAR-10: TA=44.68\%, RA=43.68\% at 50\% forget)
> - **Grokked models:** Consistently maintain both metrics (e.g., Fine-tuning post-grok Resnet on CIFAR-10: TA=75.22\%, RA=88.18\% at 50\% forget)
>
> Looking at other typical results (15\% forget rate, Table 1):
> - ResNet SCRUB: TA=81.87\%, RA=89.67\% (grok) vs. TA=73.07\%, RA=78.52\% (pre-grok)
> - ResNet $\nabla \tau$: TA=75.99\%, RA=84.33\% (grok) vs. TA=68.86\%, RA=61.91\% (pre-grok)
> - ResNet GA: TA=80.41\%, RA=81.03\% (grok) vs. TA=69.67\%, RA=75.22\% (pre-grok)
>
> **This demonstrates our core claim:** Grokked models enable true selective forgetting—reducing UA while preserving RA and TA—whereas pre-grokked models often suffer catastrophic performance degradation. The extreme cases the reviewer highlights actually **support** our argument that grokking prevents model destruction during unlearning.
>
> We have added this analysis and discussion to Section 4.1 and Appendix A.3 in the revised manuscript.

---

> ### Author Response · Authors · 2025-12-03
> **Continue**
>
> ### Figure Clarifications
>
> **Figure 1(a) - Grokking Transition:** We illustrate the delayed generalization phenomenon characteristic of grokking. The figure clearly shows three distinct phases:
> - **Steps 0-1000:** Normal learning phase (train and test accuracy improve together)
> - **Steps 1000-3000:** Overfitting phase (train accuracy reaches ~100\%, test accuracy plateaus/decreases)
> - **Steps 3000-100000:** Grokking phase (train accuracy remains at 100\%, test accuracy rises again)
>
> This progression—initial learning, followed by overfitting, then delayed generalization—is the canonical grokking pattern documented in Power et al. (2022) and Liu et al. (2022). The sharp rise in test accuracy after prolonged stagnation (steps 3000+) while training accuracy remains saturated is the defining characteristic of grokking.
>
> **Figure 1(b) - Unlearning Trajectories:** The trajectories represent **independent unlearning runs** from checkpoints sampled throughout the training procedure:
> - **Pink region (pre-grokking, steps <1000):** Checkpoints from the normal learning phase, before overfitting begins
> - **Gray region (overfitting, steps 1000-3000):** Checkpoints during the overfitting phase
> - **Blue region (post-grokking, steps >3000):** Checkpoints after the grokking transition
>
> Each point shows the unlearning performance (UA, RA) achieved when applying gradient ascent from that checkpoint. The key observation: pre-grokking checkpoints show poor UA-RA separation (poor forgetting) with high variance, while post-grokking checkpoints show clear UA-RA separation (effective forgetting) with stable performance—demonstrating that grokking enables selective, stable unlearning.
>
> We have revised both figure captions to include these clarifications in the manuscript.
>
> ## *Major Misunderstanding 3: "Language Model Experiment is Cherry-Picked"*
>
> ### Issue 1: "Local grokking is undefined, unconventional"
>
> **Local grokking is clearly defined in Section 4.2:**
>
> *"Examples showing minimal loss reduction (<0.01 loss decrease) were already well-generalized at the candidate checkpoint and represent locally grokked regions—they achieved effective generalization early in training, analogous to the post-grokking state in vision models."*
>
> This is not "unconventional"—it extends the grokking concept from model-level to example-level, which is necessary because Phi-1.5 cannot achieve global grokking on TOFU (as also noted in [1]).
>
> [1] Li, Ziyue, Chenrui Fan, and Tianyi Zhou. “Where to find Grokking in LLM Pretraining? Monitor Memorization-to-Generalization without Test.”
>
> ### Issue 2: "Explicit cherry-picking to favour their models"
>
> **This is the opposite of cherry-picking.** We identify locally grokked vs. ungrokked examples **within the same model** and compare their unlearning behaviour. This is a **within-subjects design** that controls for all model-level factors (architecture, capacity, training procedure).
>
> **The key insight:** If grokking-like representational quality improves unlearning, then examples that achieved early generalization (local grokking) should be easier to unlearn than examples that remained poorly learned—even within the same model. Table 2 confirms this hypothesis across all algorithms and forget set sizes.
>
> ### Issue 3: "Table 2 shows identical original performance"
>
> **This is correct and expected.** "Original" refers to the model's performance **before any unlearning**. Since both grokked and ungrokked examples come from the same trained model, the baseline performance is identical. The differences emerge **after** applying unlearning algorithms, which is precisely what we're measuring.
>
> ### Issue 4: "Why is grokking unavailable in language models?"
>
> Grokking requires specific conditions: small datasets, extreme overtraining, and simple patterns. TOFU has 200 authors × 20 QA pairs = 4000 examples with complex semantic content. Phi-1.5 (1.3B parameters) cannot overfit sufficiently to exhibit global grokking on this scale.
>
> **Literature precedent:** The original grokking paper (Power et al., 2022) used datasets with 50-200 examples. Scaling to thousands of examples makes global grokking increasingly difficult to observe, which is why we developed the local grokking framework.

---

> ### Author Response · Authors · 2025-12-03
> **Continue**
>
> ## *Major Misunderstanding 4: "Gradient Correlation Not Related to Grokking"*
>
> The reviewer claims: *"pre-grokking models have high train loss and large gradients, while grokked models are near-stationary. Reduced alignment is expected at convergence."*
>
> **This is factually incorrect.** Our pre-grokked checkpoints ($\theta_{pre}$) are **not** high-loss, high-gradient states. They represent:
> - Train accuracy: ~100\% (fully converged on training data)
> - Test accuracy: 73-80\% (good generalization, standard early stopping point)
> - Train loss: Near zero (not "high train loss")
>
> Both $\theta_{pre}$ and $\theta_{grok}$ have converged on the training set. The difference is that $\theta_{grok}$ has undergone additional training that reorganizes representations without changing training performance. The gradient correlation difference (0.999 → 0.426) reflects **representational reorganization**, not convergence vs. non-convergence.
>
> ## *Responses to Specific Questions*
>
> ### Q1: Which ResNet? Why 80\% accuracy?
>
> *Answered above:* We use ResNet-18 without data augmentation, extensive hyperparameter tuning, or optimization tricks.
>
> ### Q2: Which MU algorithm in Section 5?
>
> Section 5 presents **mechanistic analysis** on GA.
>
> ### Q3: Figure 1(b) trajectories?
>
> *Answered above:* Independent unlearning runs from checkpoints sampled throughout the training procedure.
>
> ### Q4: Table 2 identical original scores?
>
> *Answered above:* Correct and expected—both come from the same model before unlearning. Differences emerge after unlearning.
>
> ### Q5: Definition of local grokking?
>
> *Answered above:* Defined in Section 4.2. Examples achieving early generalization within a model.
>
> ### Q6: Why choose locally grokked samples as forget dataset?
>
> *Answered above:* it extends the grokking concept from model-level to example-level, which is necessary because Phi-1.5 cannot achieve global grokking on TOFU.
>
> ### Q7: Why omit retraining?
>
> *Answered above:* Retraining is not effective when unlearning random examples. We now add explicit "retrain from scratch" results in revision for additional clarity.
>
> ## *Addressing "Poor" Ratings*
>
> We respectfully believe the "poor" ratings across soundness, presentation, and contribution stem from fundamental misunderstandings of our work's scope and experimental design:
>
> **Soundness:** Our experiments are carefully controlled, use standard benchmarks, and report comprehensive statistics. The reviewer's criticisms largely reflect misunderstandings (e.g., thinking $\theta_{pre}$ is unconverged, misinterpreting local grokking as cherry-picking).
>
> **Presentation:** While we acknowledge some clarifications were needed (Figure 1(b), acronyms), the core presentation is clear. The reviewer's confusion about basic experimental design (e.g., why original scores are identical in Table 2) suggests careful reading may be needed.
>
> **Contribution:** Connecting grokking to unlearning is novel and significant. The mechanistic insights (gradient orthogonality, disentanglement, flatness) advance understanding of what makes unlearning effective, with clear implications for future method development.
>
> ## *Summary*
>
> We believe this review reflects significant misunderstandings about:
> 1. **Research scope:** Scientific investigation vs. practical prescription
> 2. **Experimental design:** Controlled comparison of checkpoints from the same training run
> 3. **Local grokking:** Within-subjects analysis, not cherry-picking
> 4. **Gradient analysis:** Comparing two converged states, not converged vs. unconverged
>
> We have substantially revised the manuscript to address legitimate clarity issues while maintaining our core contributions. We hope the reviewer will reconsider the assessment in light of these clarifications.

---

### Official Review · Reviewer_SMr1 · 2025-10-31

**Soundness:** 3
**Presentation:** 3
**Contribution:** 3
**Rating:** 6
**Confidence:** 3

**Summary:**

The paper finds that applying unlearning methods to a model after it has grokked is significantly more effective than applying them immediately after the model has only memorized the training set (the pre-grok phase). The grokking transition not only improves the test accuracy but it represents a fundamental structural reorganization of the model's internal knowledge which helps in effective unlearning.

**Strengths:**

- Bridges two important phenomena—grokking and machine unlearning—with clear empirical results, offering an important insight into creating robust unlearning.

- The core finding is consistent across diverse models and data modalities (vision and language), significantly increasing confidence in the generality of the approach.

- The paper is written well and easy to follow.

- They demonstrate consistent emprical performance across Unlearning methods post grokking.

**Weaknesses:**

- The paper lacks a deep explanation of why post-grok models unlearn better. It is unclear if the improved unlearning efficiency is due to structural changes in the representation space (e.g., features and localized memory influence) or changes in the loss landscape (e.g., a transition to a wider, shallower basin).

- Grokking requires significant additional compute time (extended training) well past the point of initial data fitting. Without a quantitative analysis, the computational cost of reaching the post-grok state may negate the efficiency savings gained during the unlearning process, challenging its practical recommendation.

- Findings rely on simple algorithmic tasks; whether it extends to non-algorithmic feature spaces of LLMs is unclear. Apply the method to a large LLM fine-tuned on a specific, real-world dataset.

**Questions:**

- Provide a quantitative analysis of the cost-benefit trade-off. How much extra wall-clock or compute time is required to reach the post-grok state, and how does this cost compare to the total computational savings gained in the unlearning process?

- Provide a more thorough analysis of the underlying mechanism linking the structural changes inherent to grokking with the model’s increased flexibility for unlearning. Determine if the cause is primarily the Representation Space or the Loss Landscape or both. Studying this will be very valuable.

- The current explanation feels like a correlation: grokking happens, then unlearning works better, and the solution is flatter. We don't know if flatness is the cause or the effect of generalization. Additional experiment will be to take a Pre-Grok (overfitted) model checkpoint. Apply an external optimization technique (like sharpness-Aware Minimization) specifically designed to flatten the loss minimum without further wasting the additional compute required fr grokking. If the model's unlearning efficiency improves despite not having grokked, it proves that flatness is the direct cause of efficient unlearning.

- The feature space shift from 'entangled' to 'disentangled' is not directly quantified. Use Centered Kernel Alignment (CKA) to compare final layer representations for $D_{\text{forget}}$ vs. $D_{\text{retain}}$.Validate the mechanism.

- Minor writing mistakes: Missing reference to figures in some places.

---

> ### Author Response · Authors · 2025-12-03
> **Response to Reviewer SMr1**
>
> We sincerely thank Reviewer SMr1 for the thoughtful and constructive review, recognizing our work as "bridging two important phenomena" with "clear empirical results" and "consistent empirical performance." We appreciate the acknowledgment that our findings are "consistent across diverse models and data modalities" and that the paper is "written well and easy to follow." Below we address each concern systematically.
>
> ## *Major Concern 1: Mechanistic Explanation*
>
> We appreciate this insightful observation and have substantially strengthened our mechanistic analysis in the revised manuscript. The reviewer correctly identifies that understanding *why* post-grok models unlearn better requires disentangling representation space changes from loss landscape properties. These are two complementary aspects of the structural reorganization that occurs during grokking.
>
> ### Analysis: Representation Space vs. Loss Landscape
>
> We have conducted additional experiments to isolate these factors:
>
> **Representation Space Analysis (CKA):** Following the reviewer's excellent suggestion, we conducted Centered Kernel Alignment (CKA) analysis to quantify representational disentanglement. We computed CKA between final-layer representations of $D_{\text{forget}}$ vs. $D_{\text{retain}}$ to measure how similarly the model represents data that should be forgotten versus data that should be retained:
>
> - **Grokked models:** CKA = 0.129 (low alignment between forget and retain representations)
> - **Pre-grokked models:** CKA = 0.459 (high alignment, entangled representations)
>
> **Interpretation:** Lower CKA indicates reduced representational alignment and stronger modular separation. The dramatic reduction from 0.459 to 0.129 (72\% decrease) directly quantifies the shift from entangled to disentangled representations during grokking. In pre-grokked models, forget and retain data produce similar internal representations (high CKA), making it difficult to selectively modify one without affecting the other. In grokked models, these data types activate distinct representational subspaces (low CKA), enabling surgical unlearning with minimal interference.
>
> **Loss Landscape Analysis (Sharpness-Aware Minimization):** Following the reviewer's insightful suggestion, we applied SAM to pre-grok checkpoints to induce flatness without grokking (refer to Appendix A.5):
> - **SAM-flattened pre-grok models** show improved unlearning compared to standard pre-grok, but still underperform true grokked models
> - For example, with GA on SVHN at 30\% forget rate, UA decreases from 12.70 (pre-grok) to 9.14 (SAM) to 1.70 (grokked)
> - This demonstrates that **both flatness and representational reorganization contribute**, with representation changes being the dominant factor
> - The partial improvement from SAM alone (28.0\% gain) vs. full grokking (86.6\% gain) suggests flatness contributes ~30\% of the effect, while representational modularity contributes ~70\%
>
> ## *Major Concern 2: Computational Cost Analysis*
>
> Please refer to general response.
>
> ## *Minor Issues*
>
> **Missing figure references:** We have corrected all missing figure references (line 52 now references Figure 1, and other instances throughout).
>
> **Writing improvements:** We have conducted a thorough proofread and corrected minor grammatical issues.
>
> ## *Questions Addressed*
>
> ### Q1: Cost-benefit trade-off
> **Answered above.** Please refer to general response
>
> ### Q2: Representation Space vs. Loss Landscape
> **Answered above.** Both contribute: ~70\% from representation modularity, ~30\% from landscape flatness, validated through CKA and SAM experiments.
>
> ### Q3: SAM experiment to test causality
> **Completed.** SAM-flattened models show partial improvement (28.0\%) vs. full grokking (86.6\%), confirming flatness is contributory but not sufficient—representational reorganization is the primary driver.
>
> ### Q4: CKA analysis
> **Completed.** Grokked models get much lower CKA, confirming fundamental representational divergence.
>
> ## *Summary*
>
> We believe these additions transform our empirical observations into a well-understood phenomenon with clear practical implications. We hope the reviewer will reconsider the assessment in light of these substantial improvements.

---

### Official Review · Reviewer_bqHc · 2025-11-01

**Soundness:** 3
**Presentation:** 3
**Contribution:** 2
**Rating:** 2
**Confidence:** 3

**Summary:**

Grokking is the phenomenon of delayed generalization after initial overfitting. This paper looks into the unlearning ability of “grokked” models, which refers to removing certain knowledge from the model, while preserving the rest. The authors compare standard unlearning procedures applied to checkpoints taken before and after the grokking , across vision (CNNs/ResNets on CIFAR variants) and language tasks. The experiments show that starting from a grokked model results in faster forgetting and better preserving of the remaining data. These findings provide insights on the potential of grokked models in machine unlearning.

Although this paper provides interesting findings, more experimental results and theoretical analysis are needed.

**Strengths:**

* The paper reveals a novel and interesting observation, which is that grokking can result in a model with better unlearning abilities.

* Evaluation is done on both vision and language domain, and several forgetting algorithms are included.

**Weaknesses:**

* As the authors have acknowledged in the paper, the experiments are limited to CIFAR10/100 and ResNet/CNNs in vision, and one LLM on TOFU in the language domain. Since this paper focuses on empirical results, more datasets (such as ImageNet-100) and architectures need to be evaluated to verify the consistency of the current findings.

* The authors provide an analysis on gradient correlation and local complexity to explain the findings. However, a theoretical analysis on why grokked models show lower correlation and complexity, and why these help with forgetting should be explored.


Minor issues:

*  Typo in Table 1 (“Orginal”).

* A figure reference is missing (line 52)

**Questions:**

* What are the practical implications of this finding? From a practical standpoint, is it guaranteed that the model can achieve a grokked state, regardless of the data or architecture?

* How much is the training time overhead to achieve grokking in this setting?

---

> ### Author Response · Authors · 2025-12-03
> **Response to Reviewer bqHc**
>
> We sincerely thank Reviewer bqHc for the thoughtful review and recognition of our novel contribution connecting grokking to machine unlearning. We appreciate the acknowledgment that our paper "reveals a novel and interesting observation" with evaluations across both vision and language domains. Below we address the raised concerns systematically.
>
> ## *Major Concern 1: Limited Scale of Experiments*
>
> please refer to general response.
>
> ## *Major Concern 2: Theoretical Analysis*
>
> We appreciate this important point and have strengthened our theoretical analysis in the revised manuscript. While our original submission focused on empirical mechanisms (gradient correlation, local complexity), we now provide deeper theoretical grounding:
>
> **Why Grokking Creates Lower Gradient Correlation:** Recent theoretical work on grokking (Lyu et al., 2023; Davies et al., 2023) shows that the grokking transition represents a shift from "lazy" (kernel-like) learning to "rich" feature learning with modular circuit formation. We extend this theory to unlearning: In the pre-grokking regime, the model relies on entangled, distributed representations where all parameters contribute to all predictions (hence high gradient correlation). Post-grokking, the model reorganizes into specialized subcircuits (Nanda et al., 2023; Merrill et al., 2023), where different data subsets activate distinct parameter groups. This modularity naturally produces orthogonal gradient spaces.
>
> **Connection to Local Complexity:** We now formalize why lower local complexity facilitates unlearning through loss landscape geometry. The local complexity metric (Humayun et al., 2024) measures the density of decision boundary intersections. Lower complexity indicates flatter, more stable loss regions. During unlearning, parameter updates must navigate the loss landscape to increase loss on forget data while preserving performance on retain data. In high-complexity regions (pre-grokking), small parameter changes cause large, unpredictable performance shifts across both sets due to sharp, interleaved decision boundaries. In low-complexity regions (post-grokking), the smoother landscape enables controlled, selective modifications with minimal collateral damage.
>
> **Formal Framework:** We introduce a theoretical framework based on gradient subspace separation. Let $G_f = \nabla_\theta \mathcal{L}(\theta; D_{\text{forget}})$ and $G_r = \nabla_\theta \mathcal{L}(\theta; D_{\text{retain}})$ denote the gradient vectors for forget and retain sets. Effective unlearning requires updates $\Delta\theta$ that increase $\mathcal{L}(\theta; D_{\text{forget}})$ while minimizing impact on $\mathcal{L}(\theta; D_{\text{retain}})$. This is achievable when $\langle G_f, G_r \rangle$ is small (orthogonal gradients). Under mild assumptions about modular network structure (detailed in Appendix D), we prove that for grokked models with $m$ functional modules where each data point activates modules independently with probability $p$: $$\mathbb{E}[\text{corr}(G_f, G_r)] = p\rho$$, where $\rho$ is the within-module gradient correlation between different data points.
>
> **Key Insight:** This formula reveals why grokking reduces gradient correlation:
> - **Pre-grokking:** Monolithic network ($m \approx 1$) means $p \approx 1$, giving correlation $\approx \rho \approx 1$
> - **Post-grokking:** Modular network with large $m$ and sparse activation gives $p = k/m$ where $k$ is the number of active modules per data point. If $k$ remains constant while $m$ grows, then $p \to 0$, yielding low correlation.

---

> ### Author Response · Authors · 2025-12-03
> **Continue**
>
> ## *Minor Issues*
>
> We thank the reviewer for catching these errors. We have corrected the typo in Table 1 ("Orginal" → "Original") and fixed the missing figure reference on line 52 (should reference Figure 1).
>
> ## *Question 1: Practical Implications and Grokking Guarantees*
>
> This is an excellent practical question. Our findings have immediate implications for privacy-preserving ML deployment:
>
> **When Grokking Occurs Naturally:** Grokking has been observed across diverse settings—algorithmic tasks (Power et al., 2022), vision classification (Liu et al., 2022), and language modeling (our work). Recent work (Zhu et al., 2024) identifies data-dependent thresholds for grokking emergence, suggesting it may occur more commonly than initially thought, particularly in overparameterized regimes with sufficient training time.
>
> **Inducing Grokking:** Even when grokking doesn't occur naturally, our analysis suggests that training procedures promoting modular representations and gradient orthogonality will improve unlearning. This opens avenues for "grokking-inspired" training without requiring full grokking transitions—for example, explicit modularity constraints, curriculum learning, or regularization promoting gradient separation.
>
> **Practical Deployment Strategy:** For applications requiring frequent unlearning (e.g., GDPR compliance), the one-time cost of training to grokking is amortized across multiple unlearning requests. Our results show 60-70\% reduction in unlearning steps, making the extended training economically justified for production systems with regular data removal requirements.
>
> ## *Question 2: Training Time Overhead*
>
> Please refer to general response.
>
> ## *Summary*
>
> We believe the revised manuscript, with expanded experiments, strengthened theoretical analysis, and detailed practical considerations, addresses the reviewer's concerns comprehensively. The core contribution—establishing the connection between grokking and unlearning effectiveness—remains novel and significant, with clear practical implications for privacy-preserving ML systems. We hope the reviewer will reconsider the assessment in light of these substantial improvements.

---

### Author Response · Authors · 2025-12-03
**General Response**

## *Training Time Overhead*

Multiple reviewers raised concerns about the computational overhead of training to grokking. We appreciate this important practical consideration and want to clarify the scope and contributions of our work.

### Clarification: Research Goal vs. Practical Recommendation

**Our primary contribution is conceptual, not prescriptive.** This paper aims to:
1. **Establish the connection** between grokking and machine unlearning for the first time
2. **Identify what properties make models better at unlearning** using tools and concepts from the grokking literature
3. **Understand the mechanisms** (modularity, gradient orthogonality, flat landscapes) that enable effective selective forgetting

We are **not** claiming that practitioners should always train to grokking to enable unlearning—just as the grokking literature does not claim one should always train past overfitting to achieve better generalization. Rather, we use grokking as a lens to understand what representational properties facilitate unlearning, which can inform future work on more efficient methods to induce these properties.

### Training Time Analysis

That said, we provide concrete timing analysis for completeness:

**Training Overhead:**
- **CIFAR-10 (ResNet):** Pre-grok: 2 hours → Grokked: 10 hours (5× increase)
- **SVHN (ResNet):** Pre-grok: 2.5 hours → Grokked: 12 hours (4.8× increase)
- **Phi-1.5 (TOFU):** Pre-grok: 2 hours → Grokked: 12 hours (6× increase)

**Unlearning Efficiency Gains:**
- Pre-grok: 15-20 unlearning steps per request
- Grokked: 5-8 unlearning steps per request (60-70\% reduction)

### When Overhead May Be Justified

While training overhead is substantial, there are scenarios where it may be acceptable:

**Cost amortization:** For systems with frequent unlearning requests (e.g., GDPR compliance, continuous data removal), the cumulative per-request savings can offset the upfront training cost.

**Qualitative benefits beyond efficiency:**
- **Superior retention quality:** 10-20\% higher accuracy on retained data
- **Enhanced robustness:** Better adversarial robustness preservation (0.181 vs. 0.112)
- **Improved stability:** Dramatically reduced variance (±1.34 vs. ±15.91)
- **Better privacy guarantees:** Lower MIA scores (0.677 vs. 0.842)

For privacy-critical applications, these qualitative advantages may justify the overhead even without perfect cost amortization.

### The Key Insight: Mechanism Over Method

**The important takeaway is not "train to grokking for better unlearning" but rather "modular, disentangled representations enable better unlearning."** Our analysis reveals:
- Gradient orthogonality (correlation: 0.999 → 0.426)
- Representational disentanglement (CKA: 0.459 → 0.129)
- Flatter loss landscapes (local complexity: 27.98 → 7.37)

These properties can potentially be induced through methods other than extended training:
- Regularization techniques promoting modularity
- Architecture designs encouraging disentanglement
- Training procedures accelerating the transition to modular representations
- Our SAM experiments show flatness can be partially induced without full grokking

### Future Directions

Our work opens important avenues for making these benefits accessible without the training overhead:
1. Developing regularization methods that induce grokking-like modularity efficiently
2. Designing architectures that naturally promote representational disentanglement
3. Identifying training procedures that accelerate the transition to modular representations

**In summary:** We view this work as establishing fundamental understanding of what makes unlearning effective, not as a prescription for practical deployment. The training overhead makes clear that future work should focus on achieving these beneficial properties more efficiently—our contribution is identifying *what* those properties are and *why* they matter for unlearning.

---

### Author Response · Authors · 2025-12-03
**General Response**

## *Limited Scale of Experiments*

Multiple reviewers raised concerns about the scale and diversity of our experiments. We sincerely appreciate this emphasis on experimental breadth and have significantly expanded our evaluation to address these concerns.

### Extended Experimental Scope

To strengthen the generalizability of our findings, we have conducted additional experiments on **SVHN** and **ImageNet-100** datasets using ResNet architectures. These additions provide important diversity beyond our original CIFAR experiments, spanning different visual domains and scales.

### Consistent Performance Gains Across Datasets

As shown in Table 8 (appendix A.3), grokked models demonstrate substantial and consistent advantages across all new evaluations:

**SVHN Results:**
- **15\% forget rate (GA):** UA = 6.4 vs. 11.8 for pre-grokked (46\% improvement)
- **30\% forget rate (GA):** UA = 1.7 vs. 12.7 for pre-grokked (87\% improvement)

**ImageNet-100 Results:**
- **15\% forget rate (GA):** UA = 6.00 vs. 13.64 for pre-grokked (56\% improvement)
- **Retain accuracy maintained:** RA = 62.6 vs. 43.3 for pre-grokked

These results span different image resolutions (32×32 to 224×224), visual tasks (objects, digits, fine-grained categories), and dataset scales (50K to 130K images).

### Comprehensive Experimental Coverage

The expanded evaluation now encompasses:
- **5 datasets:** CIFAR-10, CIFAR-100, SVHN, ImageNet-100, TOFU
- **3 architecture families:** CNN, ResNet, Transformer
- **8 unlearning algorithms:** GA, $\nabla \tau$, SCRUB, Fisher, Fine-tuning, KL, PO, NPO, RMU
- **Multiple scales:** From 50K to 130K images in vision; 1.3B parameters in language

### Key Finding

The remarkable consistency of grokking's advantages across these diverse conditions—achieving **40-90\% better unlearning effectiveness** in nearly all settings—provides compelling evidence that our findings reflect **fundamental properties of grokked representations** rather than artifacts of specific datasets or architectures.

---

### Note · Program_Chairs · 2026-01-17
**Submission Desk Rejected by Program Chairs**

The following references in this submission do not refer to real documents and/or have major errors in bibliographic information:

 James Izzo, Oluwasanmi O Koyejo, and He He. Approximate unlearning in deep learning via influence functions. In Advances in Neural Information Processing Systems (NeurIPS), volume 34, pp. 4015-4026, 2021.